# Measuring and Mitigating Post-Hoc Rationalization in Reverse Chain-of-Thought Generation

**Guangyue Peng** [1†] **Zongchao Chen** [2] **Wen Luo** [1] **Yuntao Wen** [3] **Wei Li** [1] **Ruixiang Feng** [3] **Ran Le** [2]
**Chen Yang** [2] **Zhenwei An** [2] **Yang Song** [2] **Tao Zhang** [2] **Houfeng Wang** [2]

## Abstract

Reverse Chain-of-Thought Generation (RCG) synthesizes reasoning traces from query-answer pairs, but it risks producing post-hoc rationalizations: when models can see the answer during generation, the answer serves as a cognitive anchor that shapes the entire explanation. We formalize this phenomenon through a three-level measurement hierarchy: lexical, entropic, and probabilistic anchoring, which capture surface artifacts, entropy dynamics, and latent answer dependence, respectively. We analyze semantic suppression, the intuitive mitigation strategy that instructs models to ignore the answer, and find that it is counterproductive: while it reduces lexical overlap, it paradoxically increases entropic and probabilistic anchoring. We attribute this failure to active monitoring of the forbidden answer, which inadvertently deepens dependence on it. To break this cycle, we propose Structural Skeleton-guided Reasoning (SSR), whose core contribution is to replace answer suppression with structural decoupling: SSR first generates a response-abstracted functional skeleton designed to limit direct answer encoding and then uses it as a structural target for full trace generation. Experiments across open-ended reasoning benchmarks show that SSR consistently mitigates anchoring, and that Distilled SSR (SSR-D), a distillation variant that internalizes skeleton-guided reasoning from teacher-generated traces, achieves up to 10% improvement over suppression baselines while mitigating out-of-distribution (OOD) degradation. Code is available at https:

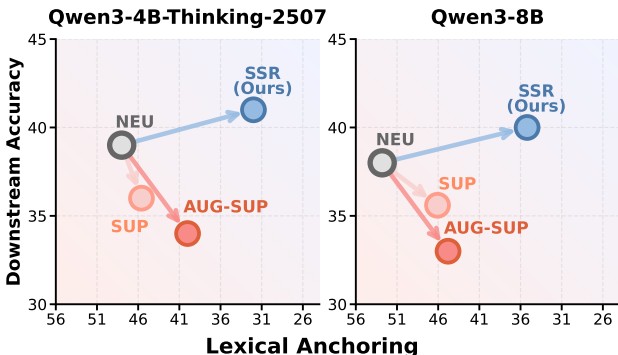

*Figure 1.* Analysis of the relationship between *Lexical Anchoring* and *Downstream Accuracy* (from Table 3). The plot reveals that lexical anchoring is a poor indicator of model utility, as semantic suppression methods (SUP/AUG-SUP) tend to "deceive" this metric, obtaining smaller anchoring while simultaneously suffering from a drop in actual downstream performance. In contrast, our SSR method achieves improvements in both dimensions.

//github.com/viniferagy/SSR.

## 1. Introduction

The effectiveness of Large Language Models (LLMs) in complex reasoning tasks depends critically on the quality of intermediate reasoning traces (Wei et al., 2022; Kojima et al., 2022; Chu et al., 2024). While expert-verified query-answer pairs are abundant (Cobbe et al., 2021; Hendrycks et al., 2021), the scarcity of corresponding step-by-step derivations creates a significant bottleneck for acquiring and transferring reasoning capabilities. Reverse Chain-of-Thought Generation (RCG) addresses this gap by synthesizing intermediate reasoning steps that logically bridge a query to a known answer (Bhagavatula et al., 2020; Zelikman et al., 2022; Li et al., 2025b).

However, RCG is susceptible to post-hoc rationalization (Cox, 2025; Jin et al., 2026). When the answer is visible during generation, models tend to rationalize backward from the conclusion rather than genuinely derive it (Turpin et al., 2023; Lanham et al., 2023; Lewis-Lim et al., 2025). Importantly, post-hoc rationalization does not necessarily

---

[†]Work done during internship at Nanbeige Lab. [1]State Key Laboratory of Multimedia Information Processing, School of Computer Science, Peking University [2]Nanbeige Lab, BOSS Zhipin [3]University of Electronic Science and Technology of China. Correspondence to: Yang Song <songyang@kanzhun.com>, Houfeng Wang <wanghf@pku.edu.cn>.

*Proceedings of the 43rd International Conference on Machine Learning*, Seoul, South Korea. PMLR 306, 2026. Copyright 2026 by the author(s).

degrade the surface quality or final accuracy of responses (Bentham et al., 2024). Rather, it undermines the reliability and utility of the reasoning traces themselves (Agarwal et al., 2024; Paul et al., 2024). Because the model has committed to the response from the outset, this pre-determined response serves as a cognitive anchor that shapes the entire explanation (Bao et al., 2025). The resulting chain-of-thought becomes less logically self-contained. An observer presented only with the query, unaware of the anchored response, would find the reasoning less accessible and coherent (Madaan et al., 2023; Arcuschin et al., 2025). This anchoring effect weakens the utility of generated traces for explainability, faithfulness verification, reasoning distillation, and reliable measurement of out-of-distribution generalization (Chua et al., 2024; David, 2025; Cetin et al., 2025).

To quantify and mitigate this phenomenon, we propose a three-level hierarchy of anchoring metrics (Lanham et al., 2023; Paul et al., 2024; Bentham et al., 2024). Beyond surface-level lexical anchoring, which simply measures token overlap between the trace and the anchored response, we examine deeper generation dynamics of the reverse chain-of-thought. We introduce entropic anchoring to capture unnatural temporal patterns in information gain, and probabilistic anchoring to measure the total mutual information between the reasoning trace and the anchored response.

We apply our measurement framework to evaluate mitigation strategies for post-hoc rationalization (Tanneru et al., 2024). The intuitive baseline is *semantic suppression*: explicitly prompting models to ignore the given response or suppress indicators of pre-determination (Cetin et al., 2025; Wang et al., 2025a). Although widely adopted as an intuitive remedy to successfully relieve the lexical anchoring, our analysis reveals that suppression fails on internal anchoring metrics. While it masks lexical overlaps, it significantly increases entropic and probabilistic anchoring, ultimately degrading the quality of reasoning traces and hurting downstream task performance (Figure 1). The model rationalizes more subtly, but more heavily. We connect this paradox with Ironic Process Theory (Wegner, 1994) from cognitive psychology: instructing a model to ignore an answer forces it to actively monitor for exclusion, thereby deepening the specific dependence it aims to cut off (illustrated in Figure 2 (b)).

To break this cycle, we propose **Structural Skeleton-guided Reasoning (SSR)**. Rather than suppressing the answer, SSR decouples the structure of reasoning from its content. Generation proceeds in two phases: (1) synthesizing a "skeleton" of functional tags (e.g., PLAN → INFR) that extracts the coarse reasoning structure behind the response while discouraging direct encoding of specific response content, and (2) using this skeleton to guide generation of the

full reasoning chain. By providing a content-neutral structural target, SSR reduces anchoring across the three-level hierarchy without relying on explicit answer-suppression instructions. Because prompted SSR can still be fragile, with models omitting tags or leaking answer details into skeletons, we further introduce **Distilled SSR (SSR-D)**. SSR-D first trains an SSR-format teacher from constructed skeleton-reasoning pairs, then uses this teacher to generate distilled SSR traces for the target model. The target model learns to generate a valid skeleton and then reconstruct the full reasoning from it, turning SSR from a prompting procedure into an internalized generation pattern.

Our contributions are:

1. **Anchoring Measurement:** We propose a comprehensive three-level framework (lexical, entropic, probabilistic) to quantify the anchoring effects of pre-committed responses in the setting of reverse chain-of-thought generation.

2. **Mechanism Analysis:** We demonstrate that semantic suppression, the intuitive mitigation strategy for post-hoc rationalization, fails due to an "ironic process" that paradoxically strengthens anchoring.

3. **Methodology:** We introduce SSR to mitigate post-hoc rationalization by decoupling reasoning structure from anchored content, and further propose SSR-D as a distillation variant that trains target models on SSR traces generated by a fine-tuned SSR teacher for more reliable structural alignment and stronger downstream gains.

## 2. Anchoring Measurement

We define *Reverse Chain-of-Thought Generation* (RCG) as follows: given a query $Q$ and a pre-committed response $A$, generate a reasoning chain

$$R = (r_1, r_2, \ldots, r_T)$$

such that $R$ constitutes a coherent derivation from $Q$ to $A$. Unlike standard chain-of-thought prompting where the answer emerges from reasoning, RCG constructs explanatory traces for predetermined conclusions.

The central question in evaluating RCG is: *To what extent does the pre-committed response $A$ anchor the generation of reasoning $R$?* We formalize this through a hierarchy of three metrics, ordered from observable surface features to latent information-theoretic properties.

### 2.1. Lexical Anchoring ($\mathcal{A}_{\text{lex}}$)

**Definition 2.1.** Lexical anchoring captures the direct appearance of response tokens within the reasoning chain. We

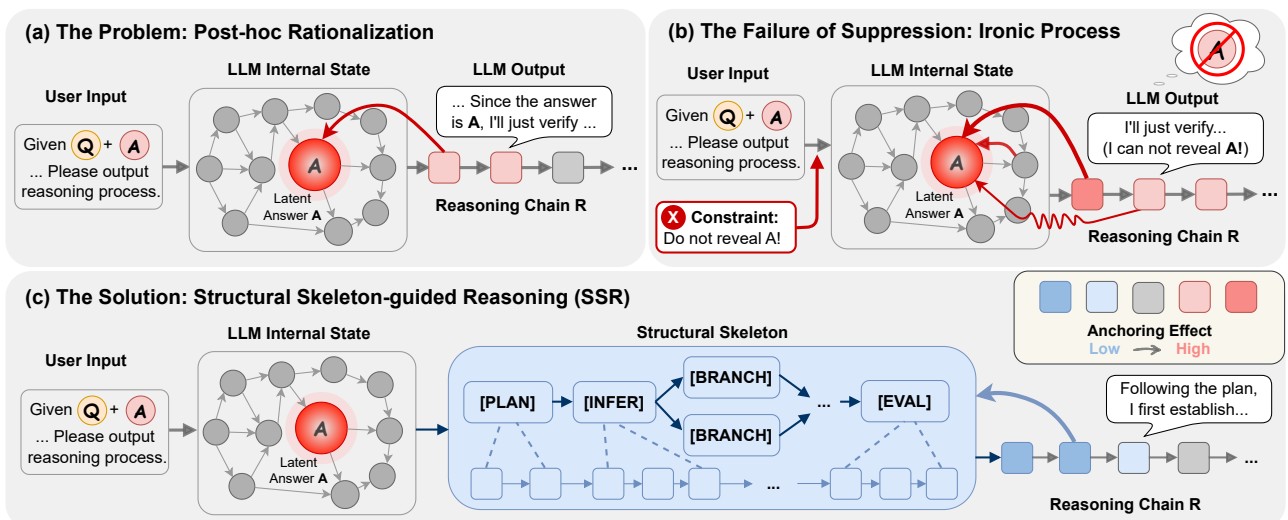

*Figure 2.* **Paradigms in Reverse Chain-of-Thought Generation.** The curved red and blue arrows indicate the anchoring effect of the pre-committed answer. (a) Post-hoc Rationalization: Visible answers cause shortcutting, resulting in rationalized reasoning chains. (b) Suppression Failure: Negative constraints trigger "ironic monitoring" of the forbidden answer, paradoxically maintaining high anchoring effect and highly rationalized chains. (c) Structural Skeleton-guided Reasoning (SSR): Decoupling content via an abstract skeleton redirects the anchoring, producing unanchored chains driven by structural information rather than answer dependency.

quantify this using ROUGE-L recall:

$$\mathcal{A}_{\text{lex}} = \frac{\text{LCS}(R, A)}{|A|} \tag{1}$$

where $\text{LCS}(R, A)$ denotes the length of the longest common subsequence between the reasoning trace and the response, and $|A|$ is the token count of the response.

While $\mathcal{A}_{\text{lex}}$ is easily minimized through paraphrasing or lexical substitution, it serves as a baseline indicator of surface-level anchoring. However, $\mathcal{A}_{\text{lex}}$ alone is an insufficient proxy for evaluating the quality of generated reasoning. As in Figure 1, reducing $\mathcal{A}_{\text{lex}}$ does not guarantee improved downstream performance, because models may simply learn to obscure the same underlying dependence.

### 2.2. Entropic Anchoring ($\mathcal{A}_{\text{ent}}$)

Beyond lexical artifacts, anchoring manifests in the dynamics of information flow. Previous studies have analyzed LLM reasoning with respect to information density. Using entropy of the predictive reasoning distribution, Gwak et al. (2025) discovers that successful LLM reasoning exhibits high local uniformity but low global uniformity. A typical CoT exhibits initial exploration with higher entropy, followed by consolidation and a final commitment phase where uncertainty collapses. Higher global variance characterizes exploration and evolution during problem solving, while local smoothness ensures coherence.

Post-hoc rationalization hinders the normal reasoning and

anchors the scope for exploration. The pre-committed response suppresses exploration, producing *high global uniformity* (artificially flat, uninformative traces), while information leakage causes *low local uniformity* (erratic transitions as the anchored answer intermittently surfaces).

Following Gwak et al. (2025), we calculate step-level information density using predictive entropy. For reasoning step $r_i$ with $M_i$ tokens and predictive distribution $p_t$ over vocabulary $\mathcal{V}$,

$$\text{ID}_i = \frac{1}{M_i} \sum_{t=1}^{M_i} H_t, \quad H_t = -\sum_{v \in \mathcal{V}} p_t(v) \log p_t(v). \tag{2}$$

**Definition 2.2.** Let $\tilde{\mathbf{u}} = [\text{ID}'_1, \ldots, \text{ID}'_N]$ be the information density vector scaled to the unit interval. Global uniformity measures exploration suppression:

$$\mathcal{G}_{\text{unif}} = \frac{1}{1 + \text{Var}(\tilde{\mathbf{u}})/\tau_g} \tag{3}$$

$\tau_g$ controls the sensitivity of $\mathcal{G}_{\text{unif}}$. Larger values make the metric less sensitive to variance differences.[1]

**Definition 2.3.** Local non-uniformity captures incoherence via the coefficient of variation of step-to-step absolute changes $\Delta_i = |\text{ID}'_i - \text{ID}'_{i-1}|$:

$$\mathcal{L}_{\text{non-unif}} = \frac{\sigma_\Delta/\mu_\Delta}{1 + \sigma_\Delta/\mu_\Delta} \tag{4}$$

---

[1]Based on preliminary experiments, we set $\tau_g = 0.1$.

**Definition 2.4.** Entropic anchoring is the geometric mean:

$$\mathcal{A}_{\text{ent}} = \sqrt{\mathcal{G}_{\text{unif}} \cdot \mathcal{L}_{\text{non-unif}}} \tag{5}$$

High $\mathcal{A}_{\text{ent}}$ indicates reasoning locked into a constrained regime, simultaneously less exploratory and less coherent. It serves as the signature of post-hoc rationalization with respect to information flow. We further discuss the theoretical foundation of $\mathcal{A}_{\text{ent}}$ in Appendix A.

### 2.3. Probabilistic Anchoring ($\mathcal{A}_{\text{prob}}$)

**Definition 2.5.** Probabilistic anchoring quantifies the extent to which the pre-committed response shapes the predictive direction of the entire reasoning chain. We measure the *bit gain rate*, the average reduction in uncertainty per response token provided by the reasoning trace, by normalizing the Pointwise Mutual Information (PMI) between $R$ and $A$:

$$\mathcal{A}_{\text{prob}} = \frac{1}{|A|} \log_2 \frac{P_\theta(A \mid Q, R)}{P_\theta(A \mid Q)}. \tag{6}$$

A high $\mathcal{A}_{\text{prob}}$ indicates that the reasoning trace $R$ reduces uncertainty about $A$ on a per-token basis. While some predictive capability is expected from valid reasoning, a disproportionately high bit gain rate suggests that the trace essentially encodes the response directly, serving as a compressed transmission channel for $A$. This metric captures anchoring that persists even when lexical and entropic artifacts are successfully masked.

## 3. Methodology

### 3.1. Baselines

**Neutral Prompting (NEU).** Our baseline employs standard chain-of-thought generation where the model is given both the query $Q$ and response $A$, then asked to produce a reasoning trace connecting them without additional constraints. This represents the default RCG setting and establishes reference anchoring levels against which mitigation strategies are compared.[2]

**Semantic Suppression (SUP).** A natural mitigation strategy instructs the model to conceal the response during generation: *"Reason step by step, but do not reveal the answer until the end."* (Wang et al., 2025a) We additionally evaluate an intensified variant (**AUG-SUP**) with stronger suppression instructions threatening the model not to disclose the relevant information (Xu et al., 2025). While intuitively appealing, we hypothesize and empirically demonstrate that this approach fails to reduce internal anchoring. Suppression preserves or amplifies response information within the reasoning dynamics despite successfully masking it in surface

text, challenging the assumption that lexical concealment equates to genuine derivation.

### 3.2. Structural Skeleton-guided Reasoning (SSR)

We propose SSR as an alternative that shifts from *suppression* to *separation*. Instead of forbidding access to the response, SSR borrows the insight from the meta-reasoning paradigm (Wang et al., 2023; Ning et al., 2024) and introduces an intermediate structural representation that decouples reasoning topology from specific semantic content in the anchored response.

We define a **Structural Skeleton**

$$S = \langle (f_1, c_1), \ldots, (f_n, c_n) \rangle$$

as a sequence of abstract steps, where each step comprises a *functional tag* $f_i$ from a closed set and a *content summary* $c_i$ describing the step's intent without revealing values (e.g., "calculate the ratio" rather than "calculate 0.5"). This abstraction is designed to encourage invariance to the specific response while preserving the logical derivation.[3]

| **Functional Tags** | | | |
| --- | --- | --- | --- |
| [PLAN] | [RETR] | [INFR] | [EVAL] |
| Planning | Retrieval | Inference | Evaluation |
| [SUMM] | [BTRK] | [RFLX] | [BRCH] |
| Summary | Backtrack | Reflection | Branch |

SSR operates via two-phase generation:

1. **Skeleton Generation:** The model generates a skeleton conditioned on query and response:

$$S \sim P(S \mid Q, A).$$

2. **Reasoning Generation:** The model generates the full reasoning trace guided by the skeleton:

$$R \sim P(R \mid S, Q, A).$$

Although $A$ remains visible in both stages, the skeleton $S$ acts as an intermediate structural target that can reduce direct response encoding during reasoning generation. By providing a content-neutral target, SSR redirects computation toward structural organization.

## 4. Experiments

We evaluate strategies for mitigating post-hoc rationalization in reverse chain-of-thought generation. Our experimental framework measures the extent to which different prompting and generation approaches reduce anchoring effects while maintaining reasoning quality.

---

[2]Detailed prompts are provided in Appendix I.

[3]Appendix C provides theoretical properties for SSR.

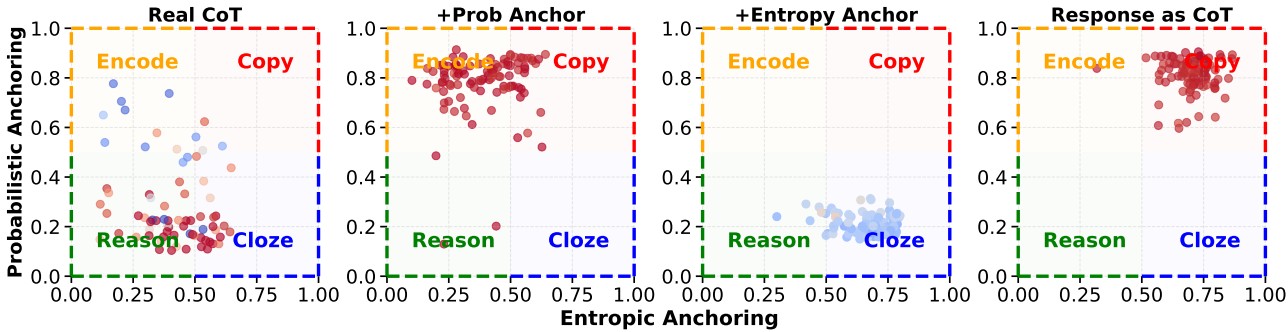

*Figure 3.* Behavioral zone construction via controlled reference conditions. **Real CoT**: standard generation without pre-committed response; **+Prob Anchor**: append response following standard CoT to induce predictive anchoring; **+Entropy Anchor**: use function-word skeleton of response to constrain exploration; **Response as CoT**: response used directly as reasoning trace. These conditions empirically locate four zones: Reason (bottom-left), Encode (top-left), Cloze (bottom-right), and Copy (top-right). Point color indicates lexical anchoring (blue: low, red: high).

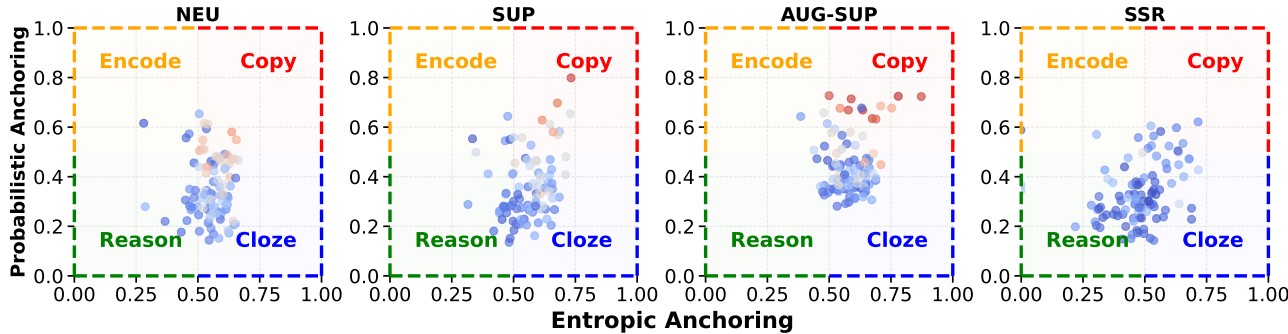

*Figure 4.* Mechanism diagnosis across generation strategies. Suppression (SUP, AUG-SUP) shifts traces among pathological zones, while SSR concentrates more traces in the Reason zone and reduces pathological spread. Point color indicates lexical anchoring (blue: low, red: high).

## 4.1. Data Construction

We sample 10,000 queries from LMArena (Chiang et al., 2024), paired with reference responses generated via a Qwen3-Max self-improvement pipeline (details in Appendix B). This setup simulates realistic reasoning distillation scenarios: high-quality responses emerge from multi-turn deliberation, but no gold-standard reasoning traces are available for supervision. Using Qwen3-4B-Thinking-2507 as target models,[4] we generate reasoning traces under each strategy and compute the anchoring metrics ($\mathcal{A}_{prob}$, $\mathcal{A}_{ent}$, $\mathcal{A}_{lex}$).

## 4.2. Behavioral Zones Construction

To empirically ground the anchoring measurement framework, we construct controlled simulations that isolate distinct behavioral regimes within the Entropic and Probabilistic Anchoring plane (Figure 3).

- **Reason (Authentic Reasoning):** The model generates

its own chain-of-thought and response without providing any pre-committed answer ($R = R^*$). This simulates standard inference where the model explores the solution space internally. Traces cluster in the bottom-left region with low anchoring on both axes.

- **Encode (Predictive Anchoring):** We append the specific response paragraphs following the reasoning trace ($R = R^* + A$). The explicit appearance of the pre-committed response anchors the model's predictive distribution, causing the trace to encode response information while maintaining normal-looking exploration dynamics. Traces populate the top-left region.

- **Cloze (Exploratory Anchoring):** We use the specific response paragraphs as the reasoning trace, with all content words masked, retaining only function words ($R = A/\{\texttt{CONTENT WORDS}\}$). The function word skeleton frames the generation path and suppresses exploratory variance without leaking substantial probabilistic information. This reduces reasoning to a "cloze" completion task. Traces cluster in the bottom-right region.

- **Copy (Severe Rationalization):** We use the specific re-

---

[4]We validate observations on Qwen3-8B in Appendix E.

*Table 1.* Anchoring metrics by generation method (Qwen3-4B-Thinking-2507). Suppression reduces $\mathcal{A}_{\text{lex}}$ but increases $\mathcal{A}_{\text{ent}}$ and $\mathcal{A}_{\text{prob}}$. SSR reduces anchoring across all axes.

| Method | $\mathcal{A}_{\text{lex}}\downarrow$ | $\mathcal{A}_{\text{ent}}\downarrow$ | $\mathcal{A}_{\text{prob}}\downarrow$ |
|---|---|---|---|
| NEU | 48.5 | 55.9 | 37.0 |
| SUP | 45.3 | 57.4 | 43.6 |
| AUG-SUP | 42.9 | 58.4 | 44.4 |
| SSR | **30.8** | **47.2** | **34.1** |
| $\Delta$ *(AUG-SUP vs. NEU)* | −11.5% | +4.5% | +20.0% |
| $\Delta$ *(SSR vs. NEU)* | −36.5% | −15.6% | −7.8% |

*Table 2.* Distribution of reasoning traces across behavioral zones (Qwen3-4B-Thinking-2507). SSR maintains the highest proportion of authentic reasoning while minimizing pathological patterns.

| Method | Reason ↑ | Encode ↓ | Cloze ↓ | Copy ↓ |
|---|---|---|---|---|
| NEU | 12.7% | **4.0%** | 54.0% | 29.3% |
| SUP | 18.2% | 9.3% | 46.6% | 25.9% |
| AUG-SUP | 5.2% | 7.6% | 44.5% | 42.7% |
| SSR | **51.2%** | 5.1% | **22.9%** | **20.8%** |

sponse paragraphs directly as the reasoning trace ($R = A$). This represents the pathological case where no reasoning process occurs. Traces occupy the top-right region with maximal anchoring on both dimensions.

### 4.3. Observations

**Suppression masks lexical anchoring but amplifies internal anchoring.** As shown in Table 1, a sharp dissociation exists between surface and latent anchoring effects. While suppression strategies (SUP, AUG-SUP) successfully reduce lexical anchoring ($\mathcal{A}_{\text{lex}}$) compared to the neutral baseline, they paradoxically increase both entropic ($\mathcal{A}_{\text{ent}}$) and probabilistic ($\mathcal{A}_{\text{prob}}$) anchoring. The intensified AUG-SUP condition exacerbates this effect: while achieving the lowest lexical anchoring among suppression methods, it produces the highest internal anchoring scores. Stronger negative constraints may encourage the model to keep a more salient internal representation of the response precisely to avoid generating it explicitly; the model rationalizes more subtly, but more heavily.

**Suppression induces pathological reasoning.** Visualizing reasoning traces in the Entropic and Probabilistic Anchoring plane reveals how suppression distorts the reasoning process (Figure 4). Under NEU, traces already show substantial mass in pathological zones, indicating that default RCG is vulnerable to anchoring. Suppression (SUP, AUG-SUP) does not reliably move traces into authentic reasoning: it shifts mass among pathological regimes, increasing Encode behavior and, under AUG-SUP, producing the largest Copy fraction. The model trades genuine derivation for constrained exploration or disguised encoding even when surface copying is reduced.

**SSR achieves consistent reduction across all metrics.** Unlike suppression methods, which trade lower lexical overlap for higher latent anchoring, SSR consistently reduces anchoring across all three levels (Table 1). By grounding generation in a pre-planned structural skeleton, SSR achieves the lowest lexical, entropic, and probabilistic anchoring scores among all methods. The reduction in lexical anchoring demonstrates effective surface-level mitigation, while

the reductions in entropic and probabilistic anchoring indicate that generated traces exhibit more natural information dynamics and greater independence from the pre-committed response.

**SSR shifts traces toward more self-contained reasoning dynamics.** Analyzing the behavioral zone distributions (Figure 4), SSR counteracts the pathological shifts induced by suppression. While suppression leaves most traces in Encode, Cloze, or Copy zones, SSR achieves over half the traces in authentic reasoning (Table 2). The structural skeleton helps avoid aimless generation by providing a plan and can discourage rationalization by defining granular step-wise intents. In the SSR panel, traces shift toward the bottom-left quadrant with reduced spread into pathological zones, indicating that SSR's structural guidance improves reasoning trajectories rather than merely shifting the mean while preserving high variance.

### 4.4. Interpretation

The divergent behaviors of suppression-based methods and SSR can be understood through the lens of *Ironic Process Theory* from cognitive psychology (Wegner, 1994). When instructed to suppress a concept (e.g., "do not think of a white bear"), individuals paradoxically experience *increased* accessibility of that concept. This occurs because suppression requires an active monitoring process to detect and exclude the forbidden content, a process that necessarily keeps the target concept cognitively salient.

We argue that an analogous phenomenon manifests in LLMs under semantic suppression. It is illustrated in Figure 2 (b) that when prompted to "not reveal the answer," the model is likely to maintain a representation of what constitutes the answer to evaluate whether each generated token violates the constraint. This monitoring induces more anchoring effect, biasing the entire generation trajectory, even when the model successfully avoids surface-level copying.

SSR circumvents this paradox through a fundamentally different mechanism. Rather than imposing negative constraints that require answer monitoring, SSR provides a positive structural target that redirects the model's generative focus. As shown in Figure 2 (c), the skeleton specifies

what operations to perform (e.g., `[PLAN]` → `[BRCH]` → `[EVAL]`) without encoding what results to obtain.

The skeleton possesses a rich reasoning structure, including non-linearity and diversity (encouraged by tags such as `[BTRK]` and `[RFLX]`), which preserves room for exploration and entropy robustness, thereby mitigating the entropic anchoring effect. Meanwhile, this response-abstracted scaffold can reduce direct probabilistic anchoring by shifting generation toward structural guidance rather than monitoring forbidden content, weakening the ironic cycle that suppression can create.

# 5. Downstream Performance

We next test the effectiveness of reverse CoT under different anchoring influences, addressing three questions: (1) Do reduced anchoring effects translate to improved downstream performance? (2) Does rationalization mitigation enable better out-of-distribution generalization? (3) Does distillation strengthen structural alignment beyond prompting?

## 5.1. Experimental Setup

**Methods.**

- We continue to evaluate the strategies defined in Section 3: **NEU** (neutral prompting), **SUP** and **AUG-SUP** (semantic suppression baselines), **SSR** (structural skeleton-guided generation).

- **SSR-D** (distilled SSR). While SSR can be implemented via prompting, adherence to the structural format is often inconsistent, since models may omit tags or leak results into skeletons. SSR-D addresses this by fine-tuning the target model on SSR traces generated by a fine-tuned SSR teacher. Given a teacher-generated pair $(S^*, R^*)$, the target model is trained with two objectives: skeleton generation $\mathcal{L}_S = -\log p_\theta(S^* \mid Q, A)$ and reasoning reconstruction $\mathcal{L}_R = -\log p_\theta(R^* \mid Q, A, S^*)$. The combined objective $\mathcal{L} = \mathcal{L}_S + \mathcal{L}_R$ encourages both skeleton validity and skeleton-conditioned reasoning. Training details are provided in Appendix G.

**Data.** We sample 100k queries from LMArena (Chiang et al., 2024) and generate reference answers using Qwen3-Max. Given $(Q, A)$ pairs lacking intermediate reasoning, we generate reverse CoT using Qwen3-235B-Instruct-2507 (Yang et al., 2025a) under each baseline condition, enabling evaluation at frontier-model scale. For SSR-D, a fine-tuned Qwen3-235B-Instruct-2507 SSR teacher serves as the generator of the distilled SSR traces.

**Models.** We fine-tune Qwen3-8B and Qwen3-32B (Yang et al., 2025a) as representative open-weight thinking mod-

els, Qwen3-4B-Thinking-2507 for reasoning-specialized evaluation, and NBG4-3B-Base (Yang et al., 2025b) to test cross-family transfer.

**Benchmarks.** We evaluate across diverse reasoning domains: open-ended reasoning **ArenaHard-v2.0** (Li et al., 2025c), human-aligned emotional intelligence reasoning **EQ-Bench 3** (Paech, 2023), strict constraint adherence **IFEval** (Zhou et al., 2023b), and multi-turn instruction-following **MultiChallenge** (Deshpande et al., 2025). For out-of-distribution evaluation, we use **GPQA-Diamond** (Rein et al., 2023) for scientific reasoning and **AIME 2025** (Zhang & Math-AI, 2025) for difficult mathematical reasoning. All evaluation details can be found in Appendix F.

## 5.2. Main Results

Table 3 presents in-distribution results across four benchmarks and model scales.

**Reduced anchoring yields consistent performance gains.** SSR-D achieves the highest scores across all reported benchmarks and model scales, with consistent gains across model scales/families. On ArenaHard, improvements over NEU reach +9.1 points for Qwen3-4B-Thinking-2507, while MultiChallenge shows consistent gains up to +3.1 points. This correlation between lower internal anchoring (Table 1) and higher downstream accuracy supports the usefulness of our framework: structured reasoning traces provide stronger training signals than post-hoc rationalizations.

**Suppression degrades multi-turn reasoning.** While SUP and AUG-SUP yield marginal single-turn improvements on ArenaHard, they consistently degrade MultiChallenge performance. This asymmetry is most pronounced for Qwen3-8B, where AUG-SUP underperforms NEU by 5 points despite appearing less anchored by $\mathcal{A}_{\text{lex}}$ (Figure 1). The pattern confirms that suppression-induced traces lack global coherence for sustained multi-turn reasoning, consistent with elevated $\mathcal{A}_{\text{ent}}$ in Table 1.

**Distillation amplifies structural alignment.** Comparing SSR and SSR-D reveals that prompting alone captures roughly half the potential benefit (e.g., +5 vs. +9 on ArenaHard for Qwen3-8B). The consistent SSR-D advantage reflects learned internalization of the skeleton pattern, ensuring reliable format adherence and stronger rationalization mitigation. Cross-architecture transfer to NBG4-3B-Base further suggests that the benefit is not limited to the Qwen model family.[5]

---

[5]We ablate the components of SSR in Appendix H.

*Table 3.* Main results across core benchmarks. Best results are in **bold**, and second-best are underlined. SSR and SSR-D consistently outperform suppression-based methods, with SSR-D achieving the largest gains. Suppression methods show marginal improvement on ArenaHard but degrade multi-turn performance. (Arena: ArenaHard, EQ: EQ-Bench 3, IF: IFEval, MC: MultiChallenge)

| Method | Qwen3-8B-Think | | | | Qwen3-32B-Think | | | | Qwen3-4B-Think-2507 | | | | NBG4-3B-Base | | | |
|---|---|---|---|---|---|---|---|---|---|---|---|---|---|---|---|---|
| | Arena | EQ | IF | MC | Arena | EQ | IF | MC | Arena | EQ | IF | MC | Arena | EQ | IF | MC |
| NEU | 50.8 | 76.4 | 80.0 | 38.2 | 68.2 | 85.5 | 83.4 | 43.8 | 41.7 | 79.3 | 76.7 | 38.9 | 32.4 | 79.6 | 74.3 | 34.5 |
| SUP | 51.5 | 77.5 | 81.2 | 35.5 | 68.8 | 85.6 | 84.5 | 41.5 | 42.3 | 79.2 | 78.5 | 36.6 | 32.9 | 81.5 | 76.5 | 33.4 |
| AUG-SUP | 52.8 | 78.3 | 81.5 | 33.0 | 69.1 | 86.4 | 83.2 | 41.3 | 43.5 | 79.0 | 80.1 | 34.5 | 33.2 | 82.8 | 77.2 | 33.0 |
| SSR | 56.3 | 82.6 | 82.9 | 39.4 | 70.1 | 87.7 | 84.8 | 44.6 | 46.1 | 81.9 | 81.6 | 40.2 | 34.6 | 84.4 | 77.6 | 35.8 |
| SSR-D | **59.5** | **86.1** | **83.7** | **41.2** | **72.0** | **89.6** | **85.1** | **45.0** | **50.8** | **83.1** | **83.8** | **42.0** | **37.2** | **86.5** | **78.2** | **37.1** |

*Table 4.* Out-of-distribution performance on GPQA-Diamond and AIME 2025. **Bold** and underlined values denote the best and second-best results among trained methods (excluding w/o Train).

| Method | Qwen3-8B-Think | | Qwen3-32B-Think | |
|---|---|---|---|---|
| | GPQA-D | AIME | GPQA-D | AIME |
| w/o Train | 58.1 | 70.0 | 64.1 | 73.3 |
| NEU | 49.5 | 33.3 | 60.6 | 56.7 |
| SUP | 51.0 | 35.0 | 60.1 | 52.5 |
| AUG-SUP | 51.2 | 33.3 | 61.1 | 49.2 |
| SSR | 53.2 | 40.0 | 62.7 | 56.7 |
| SSR-D | **56.6** | **44.2** | **65.2** | **59.2** |

*Table 5.* Filtering and teacher-model validation on 16k subsets. The Qwen3-Max-Preview setting compares against standard forward-reasoning rollout filtering; the Kimi2.5 setting repeats data construction and evaluation with a different teacher/evaluator. Higher is better.

| Method | Qwen3-Max-Preview | Kimi2.5 |
|---|---|---|
| Native rollout | 45.2 | 37.8 |
| Best-of-3 rollout | 46.7 | 39.2 |
| NEU | 44.0 | 35.4 |
| SUP | 45.5 | 36.9 |
| AUG-SUP | 45.8 | 37.5 |
| SSR | 48.2 | 39.6 |
| SSR-D | **55.1** | **44.0** |

## 5.3. Out-of-Distribution Generalization

Table 4 reveals that standard RCG training can substantially degrade OOD performance, especially on AIME 2025. NEU-trained Qwen3-8B falls to less than half its untrained baseline, as the training data lack mathematical content comparable to competition-level problems. Suppression methods provide no recovery, while AUG-SUP even worsens AIME for Qwen3-32B by 7.5 points.

While all methods suffer from this domain gap, SSR and SSR-D substantially outperform suppression baselines through anchoring mitigation. For Qwen3-8B, SSR-D recovers about 30% of the AIME gap to the untrained baseline, while Qwen3-32B with SSR-D exceeds its untrained GPQA-Diamond score. We attribute this advantage to structural decoupling: the model acquires transferable derivation patterns rather than content-specific shortcuts. Training data coverage determines the ceiling of OOD performance; anchoring mitigation determines how much of that ceiling is preserved.

## 5.4. Additional Validation of RCoT Quality

**Filtering baselines and teacher-model bias.** To compare against a strong forward-reasoning baseline, we evaluate a 16k subset on ArenaHard, where Qwen3-Max-Preview generates $r$ standard reasoning rollouts ($r = 3$) and selects the best among them by self-judging using the same teacher as judge. When high-quality answers are available, ordinary RCG is not automatically stronger than repeated rollouts

from a strong reasoning model: Best-of-3 filtering improves the base generator from 45.2 to 46.7, while NEU reaches 44.0. In contrast, SSR reaches 48.2 and SSR-D reaches 55.1 (Table 5), showing that structurally guided RCG can better leverage the given answer while preserving reasoning ability. To test whether the conclusion depends on Qwen3-Max as the sole teacher/evaluator, we repeat the same subset setting with Kimi2.5; SSR and SSR-D again outperform NEU, SUP, and AUG-SUP. This demonstrates that the gains are not specific to one teacher family.

**External metric validation and qualitative evidence.** A 500-sample LLM-as-judge study provides convergent external validation. In the *self-contained derivation* pass, the judge sees only the query and reasoning trace and scores whether the trace can stand on its own. In the *post-hoc dependence* pass, the judge additionally sees the pre-committed answer and scores how answer-driven the trace appears. Two independent judges, Claude Opus 4.6 (claude-opus-4-6, Anthropic, 2026) and GPT-5.4 (gpt-5.4-2026-03-05, OpenAI, 2026), consistently rank SSR-D best on self-contained derivation and lowest on post-hoc dependence, with SSR also ahead of SUP and NEU (Tables 6 and 7). Appendix D reports a qualitative case study and structural analysis. In both tables, Var. denotes sample variance over the 500 judged examples.

*Table 6.* LLM-as-judge validation with Claude Opus 4.6. Higher self-contained derivation and lower post-hoc dependence are better.

| Method | Deriv. ↑ | Var. | Dep. ↓ | Var. |
|--------|----------|------|--------|------|
| NEU | 4.375 | 0.890 | 3.563 | 1.217 |
| SUP | 4.439 | 0.929 | 3.592 | 1.028 |
| SSR | 4.615 | 0.492 | 3.448 | 0.776 |
| SSR-D | **4.744** | 0.263 | **2.698** | 0.684 |

*Table 7.* LLM-as-judge validation with GPT-5.4. Higher self-contained derivation and lower post-hoc dependence are better.

| Method | Deriv. ↑ | Var. | Dep. ↓ | Var. |
|--------|----------|------|--------|------|
| NEU | 3.580 | 1.458 | 4.480 | 0.919 |
| SUP | 3.690 | 1.529 | 4.380 | 1.208 |
| SSR | 4.113 | 0.768 | 4.155 | 1.445 |
| SSR-D | **4.410** | 0.366 | **3.040** | 1.312 |

## 6. Related Work

**Reverse Chain-of-Thought Generation.** While Chain-of-Thought (CoT) enables systematic reasoning (Wei et al., 2022; DeepSeek-AI, 2025; Peng et al., 2025a; Feng et al., 2026), supervised data often lacks intermediate traces. Existing solutions employ iterative self-improvement (Zelikman et al., 2022), reinforcement learning (Shao et al., 2024), or trajectory reconstruction (Shridhar et al., 2023; Li et al., 2025b; Wang et al., 2025a). However, these methods expose the target answer during trace generation, inducing anchoring effects that compromise reasoning quality. We address this through structural decoupling, reducing direct dependence on answer-specific content.

**Post-hoc Rationalizations in Chain-of-Thought.** LLM rationales are not always faithful (Lanham et al., 2023; Paul et al., 2024; Tanneru et al., 2024). When models decide on a response before starting to reason, their post-hoc rationalizations may be influenced by unstated biases (Turpin et al., 2023; Lyu et al., 2023) or contain deceptive shortcuts (Li et al., 2024; Yee et al., 2024; Bentham et al., 2024). Recent answer-attribution work further shows that LRM answers can arise from competing reasoning and retrieval mechanisms, and that retrieval-dominant behavior can produce post-hoc explanations for memorized answers (Wang et al., 2026). While current mitigation methods focus on verification or process rewards (Lightman et al., 2024; Wang et al., 2025c; Feng et al., 2025), we frame the problem through an anchoring lens: answer visibility anchors generation as justification rather than derivation. We propose a metric hierarchy to quantify this phenomenon and achieve rationalization mitigation by construction through structural skeleton.

**Structural Approaches in Reasoning.** Prior work has established the importance of reasoning structure in LLM chain-of-thought (Madaan et al., 2023; Li et al., 2025a; Wang et al., 2025b; Peng et al., 2025b). Structured reasoning has been obtained via non-linear exploration (Yao et al., 2023; Besta et al., 2024), stage decomposition (Wang et al., 2023; Zhou et al., 2023a; Wen et al., 2025), and meta-reasoning (Wang et al., 2024; Zhang et al., 2025). Recent long-CoT studies also show that fully constructed trajectories can underperform emergent teacher traces in standard forward-reasoning settings (Yang et al., 2025c). This is complementary to our finding rather than contradictory: SSR is not a hand-constructed full trajectory, but a lightweight response-abstracted scaffold for the answer-visible reverse-CoT setting. Our SSR leverages structure to redirect generation away from anchoring cues. Crucially, we constrain skeletons to avoid direct answer leakage, distinguishing our approach from efficiency-driven methods such as Ning et al. (2024).

## 7. Conclusion

In this work, we formalized post-hoc rationalization in the setting of Reverse Chain-of-Thought Generation (RCG), where visible pre-committed responses drive models toward rationalization rather than derivation. Our analysis revealed that standard semantic suppression paradoxically exacerbates this issue via an *ironic process*, where the load of monitoring forbidden answers increases latent information leakage. We introduced Structural Skeleton-guided Reasoning (SSR), a paradigm that shifts from negative constraints to positive structural decoupling. By generating a response-abstracted functional skeleton, SSR provides a content-neutral structural target that can weaken the cycle of ironic monitoring. Experiments show that SSR reduces overall rationalization, while its applications in prompting and distillation consistently improve downstream task performance and improve out-of-distribution robustness in our evaluations. These results suggest that faithful reasoning may not be achieved by suppressing the undesirable, but by scaffolding the reasoning process to render shortcuts unnecessary.

## Acknowledgements

This work was supported by the National Natural Science Foundation of China (62576010) and the Academic Research Projects of Beijing Union University (NO. ZK10202405).

## Impact Statement

This paper presents work whose goal is to advance the field of machine learning by improving the measurement and mitigation of post-hoc rationalization in generated reasoning traces. More faithful reasoning traces may benefit

model evaluation, debugging, and reasoning distillation by reducing misleading explanations that merely justify pre-committed answers. The methods and metrics introduced here could also be misused to make model explanations appear more reliable than they are if reported without appropriate validation. We therefore encourage practitioners to pair anchoring metrics with task-specific correctness, robustness, and human evaluation when deploying reasoning-generation systems.

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

# A. Details of Anchoring Metrics

## A.1. Entropic Anchoring: Theoretical Foundations

### A.1.1. CONNECTION TO HUMAN COGNITIVE PROCESSING

A natural path toward understanding the reasoning capabilities of LLMs is to probe their relationship with human cognitive processing. We ground our entropic anchoring metric in the psycholinguistic hypothesis of Uniform Information Density (UID), which proposes that speakers distribute information as evenly as possible to balance clarity and efficiency (Levy & Jaeger, 2006; Meister et al., 2021).

The UID hypothesis conceptualizes language as a signal transmitted through a noisy channel with limited capacity (Meister et al., 2021; Tsipidi et al., 2024). Under this framework, speakers aim to convey information efficiently without overwhelming the listener's processing resources. When information density becomes uneven, communication deteriorates. This principle has been extensively validated in human language production and comprehension.

### A.1.2. FROM HUMAN COMMUNICATION TO LLM REASONING

Gwak et al. (2025) recently revisited the UID hypothesis in the context of LLM reasoning, asking whether step-level uniformity reflects reasoning quality. Their key insight is that while human communication benefits from global uniformity to avoid overwhelming listeners (Bhambri et al., 2025), effective LLM reasoning exhibits a fundamentally different pattern: high local uniformity but low global uniformity. Following Gwak et al. (2025), we use entropy of the predictive distribution as a proxy for information density.

# B. Iterative Self-Refinement for High-quality Answer Construction

We construct our experiment dataset using a self-improvement pipeline to simulate the situations that the high-quality responses are accessible but human-written or gold intermediate reasoning traces are lacking.

**Data source.** We sample user queries from the LMArena human preference corpus (140k conversations) hosted at `https://huggingface.co/datasets/lmarena-ai/arena-human-preference-140k` and introduced in prior work on the text arena (Chiang et al., 2024).

**Overview.** For each sampled query, we use Qwen3-Max (Yang et al., 2025a) to construct a high-quality reference answer via a multi-stage, iterative refinement pipeline. The procedure alternates between (i) generating diverse candidate responses, (ii) self-evaluating candidates along key quality dimensions, and (iii) aggregating the strongest components into improved candidates, repeating this process for multiple refinement loops.

**Pipeline stages.** Let $N$ denote the number of independent rollouts produced per query, $K$ the number of candidate slots maintained after aggregation, $M$ the number of evaluated candidates sampled for synthesizing each slot, and $T$ the number of refinement loops.

1. **Candidate generation.** We prompt Qwen3-Max to produce $N$ independent response rollouts for the same query to encourage diversity in reasoning paths and content coverage.

2. **Candidate evaluation.** The model then self-evaluates each rollout, assigning a scalar score based on accuracy, coherence, and completeness. These scores are used for quality reference during the aggregation step.

3. **Candidate aggregation.** We construct a new set of $K$ candidates. For each slot, we randomly sample $M$ evaluated candidates and synthesize an improved response by combining their highest-scoring components (e.g., correct facts, clearer explanations, or more complete coverage).

4. **Improvement loop.** The newly synthesized $K$ candidates are fed back into the evaluation and aggregation stages. We repeat this improvement cycle for $T$ loops, yielding a final, polished reference answer at convergence.

**Algorithmic description.** Algorithm 1 summarizes the refinement procedure used to produce the definitive reference answer per query.

---

**Algorithm 1** Iterative self-refinement for reference answer construction

---

**Require:** Query $x$; model $\mathcal{M}$ (Qwen3-Max); rollouts $N$; slots $K$; sample size $M$; loops $T$

1: $\mathcal{C} \leftarrow \{\mathcal{M}(x)\}_{i=1}^{N}$          {Generate $N$ independent candidates}
2: **for** $t = 1, \ldots, T$ **do**
3:     $\mathcal{S} \leftarrow \{\text{SCORE}(c) \mid c \in \mathcal{C}\}$        {Self-evaluate on accuracy/coherence/completeness}
4:     $\mathcal{C}' \leftarrow \emptyset$
5:     **for** $j = 1, \ldots, K$ **do**
6:        Sample $\{c_{j,1}, \ldots, c_{j,M}\}$ from $\mathcal{C}$ (optionally biased by $\mathcal{S}$)
7:        $\mathcal{S}_j \leftarrow \{\text{SCORE}(c_{j,m})\}_{m=1}^{M}$        {Use only scores of sampled candidates}
8:        $c_j' \leftarrow \text{SYNTHESIZE}(\{c_{j,1}, \ldots, c_{j,M}\}, \mathcal{S}_j)$
9:        $\mathcal{C}' \leftarrow \mathcal{C}' \cup \{c_j'\}$
10:    **end for**
11:    $\mathcal{C} \leftarrow \mathcal{C}'$        {Update candidates for next loop}
12: **end for**
13: **return** $\arg\max_{c \in \mathcal{C}} \text{SCORE}(c)$        {Final reference answer}

---

**Rationale.** Our construction yields reference responses that reflect the outcome of substantial deliberation and iterative error-correction, while keeping intermediate reasoning implicit. As a result, the final answer provides a strong target without exposing step-by-step traces that could be trivially copied. This design matches realistic deployment conditions, where systems are typically evaluated on inputs and final outputs rather than on access to the full internal decision process. It also creates a controlled setting in which a model must justify or explain an already-produced outcome using only the surface form of the response, which is exactly where post-hoc rationalization is most likely to arise and most difficult to detect.

## C. Theoretical Analysis of SSR

We provide an information-theoretic framework for the *skeleton-mediated* channel of Structural Skeleton-guided Reasoning (SSR). The goal of this appendix is deliberately limited: we bound the information about the pre-committed response $A$ that can be transmitted through the skeleton $S$. We do not claim a formal bound on the residual reasoning-generation channel $I(A; R \mid Q, S)$, since the second phase still conditions on $A$. The overall reduction of probabilistic anchoring is therefore an empirical result, measured by $\mathcal{A}_{\text{prob}}$ in the main experiments.

### C.1. Skeleton Capacity and Functional Invariance

Let a structural skeleton

$$S = \langle (F_i, C_i) \rangle_{i=1}^{N}$$

consist of functional tags $F_i \in \mathcal{F}$ and content summaries $C_i$. All information quantities in this appendix are measured in bits. We first work in the fixed-length setting $N = n$, and later recover the variable-length case by marginalizing over $N$. Under $N = n$, let

$$S_{<i} = \big((F_1, C_1), \ldots, (F_{i-1}, C_{i-1})\big)$$

denote the autoregressive skeleton history before step $i$, and let

$$H_i^{(n)} = (Q, N = n, S_{<i}, F_i)$$

be the random context used in the fixed-length analysis for the $i$-th content summary. Conditioning on $N = n$ is an analytical convention; it does not assert that the model observes the final length during generation.

**Definition C.1** (Sequential $\epsilon$-Functional Invariance). A skeleton generator is *sequentially $\epsilon$-functionally invariant* under fixed length $N = n$ if each content-summary step satisfies

$$I(A; C_i \mid H_i^{(n)}) \leq \epsilon_i^{(n)}.$$

Here $\epsilon_i^{(n)} \in \mathbb{R}_{\geq 0}$ is a scalar conditional-MI bound after marginalizing over the random context $H_i^{(n)}$, so these bounds compose additively under the chain rule below. We use this average mutual-information form unless stated otherwise.

A stronger pointwise sufficient condition is that, for all $(h_i, a)$ in the support of $(H_i^{(n)}, A)$,

$$D_{\mathrm{KL}}^{(2)}\Big(P(C_i \mid H_i^{(n)} = h_i, A = a) \,\|\, P(C_i \mid H_i^{(n)} = h_i)\Big) \leq \epsilon_i^{(n)},$$

where $D_{\mathrm{KL}}^{(2)}$ uses base-2 logarithms. This pointwise condition implies Definition C.1, but is stronger than the average conditional-MI requirement. Although $\epsilon_i^{(n)}$ is a theoretical primitive, it can be empirically diagnosed through held-out estimates of the pointwise bit gain $\log_2[P(C_i \mid H_i^{(n)}, A)/P(C_i \mid H_i^{(n)})]$, or by probing summaries for $A$-relevant features.

**Proposition C.2** (Skeleton-Channel Bound). *For a fixed-length $n$-step skeleton with discrete functional tags $F_i \in \mathcal{F}$, if the content summaries satisfy sequential $\epsilon$-functional invariance, then*

$$I(A; S \mid Q, N = n) \leq \sum_{i=1}^{n} H(F_i \mid Q, N = n, S_{<i}) + \sum_{i=1}^{n} \epsilon_i^{(n)} \leq n \log_2 |\mathcal{F}| + \sum_{i=1}^{n} \epsilon_i^{(n)}.$$

*For variable-length skeletons,*

$$I(A; S \mid Q) = I(A; N \mid Q) + \mathbb{E}_{n \sim N}[I(A; S \mid Q, N = n)]$$

*where each fixed-length term is evaluated under the conditional law given $N = n$, and hence*

$$I(A; S \mid Q) \leq I(A; N \mid Q) + \mathbb{E}_{n \sim N}\left[n \log_2 |\mathcal{F}| + \sum_{i=1}^{n} \epsilon_i^{(n)}\right].$$

*Proof.* For fixed $N = n$, apply the chain rule in the natural autoregressive skeleton order:

$$I(A; S \mid Q, N = n) = \sum_{i=1}^{n} I(A; F_i, C_i \mid Q, N = n, S_{<i}).$$

Expanding each pair $(F_i, C_i)$ gives

$$I(A; S \mid Q, N = n) = \sum_{i=1}^{n} I(A; F_i \mid Q, N = n, S_{<i}) + \sum_{i=1}^{n} I(A; C_i \mid Q, N = n, S_{<i}, F_i).$$

The tag terms are bounded by conditional entropy:

$$I(A; F_i \mid Q, N = n, S_{<i}) \leq H(F_i \mid Q, N = n, S_{<i}) \leq \log_2 |\mathcal{F}|.$$

The content-summary terms are bounded directly by Definition C.1, since $H_i^{(n)} = (Q, N = n, S_{<i}, F_i)$:

$$I(A; C_i \mid Q, N = n, S_{<i}, F_i) = I(A; C_i \mid H_i^{(n)}) \leq \epsilon_i^{(n)}.$$

Summing over $i$ proves the fixed-length bound. For the variable-length case, $N$ is a deterministic function of the skeleton $S$, so the chain rule gives

$$I(A; S \mid Q) = I(A; N, S \mid Q) = I(A; N \mid Q) + I(A; S \mid Q, N).$$

Equivalently,

$$I(A; S \mid Q, N) = \mathbb{E}_{n \sim N}[I(A; S \mid Q, N = n)].$$

Applying the fixed-length bound for each realized length yields the stated result. □

*Remark* C.3 (No free lunch in invariance). The useful operating regime is not $\epsilon_i^{(n)} \to 0$ at all costs. If the skeleton transmits no answer-relevant guidance through tag order, length, or abstract operations, it may become too generic to support the intended reasoning path, pushing answer dependence back into the residual term $I(A; R \mid Q, S)$ during reasoning generation. SSR therefore aims for a low-bandwidth structural channel: answer-relevant guidance is routed through coarse structure, while concrete result leakage in content summaries is discouraged.

### C.2. Connection to the Anchoring Measurement Hierarchy

**Probabilistic anchoring.** Probabilistic anchoring $\mathcal{A}_{\text{prob}}$ measures a per-response-token bit gain:

$$\mathcal{A}_{\text{prob}}(q, r, a) = \frac{1}{|a|} \log_2 \frac{P_\theta(a \mid q, r)}{P_\theta(a \mid q)}.$$

When the scoring model matches the joint distribution of $(Q, A, R)$, $\mathbb{E}[|A|\mathcal{A}_{\text{prob}}]$ corresponds to $I(A; R \mid Q)$; for comparable answer lengths, $\mathbb{E}[\mathcal{A}_{\text{prob}}]$ is therefore a length-normalized proxy for this total conditional mutual information.

Under SSR's two-phase generation, the reasoning trace is generated as

$$R \sim P(R \mid S, Q, A).$$

The dependence between $A$ and $R$ admits the bound

$$I(A; R \mid Q) \leq I(A; S \mid Q) + I(A; R \mid Q, S).$$

The first term is the skeleton-mediated channel bounded by Proposition C.2. The second term is residual anchoring during reasoning generation. In the worst case, it can be as large as $H(A \mid Q, S) \leq H(A \mid Q)$, which can make the overall bound vacuous when the skeleton provides little useful structure. Proposition C.2 therefore gives a guarantee only on information transmitted through $S$; whether total probabilistic anchoring decreases under SSR is an empirical claim, supported by Tables 1 and 11 and Figure 4.

**Entropic anchoring.** The structural skeleton specifies a reasoning topology through functional tags (e.g., [PLAN] $\rightarrow$ [BRCH] $\rightarrow$ [EVAL]). Exploration tags such as [BRCH] and [RFLX] can create natural entropy peaks, while [EVAL] and [SUMM] can mark consolidation phases. This provides a qualitative mechanism for reducing the over-uniform or erratic information-density patterns associated with post-hoc rationalization, but it is not a formal guarantee on $\mathcal{A}_{\text{ent}}$.

**Lexical anchoring.** Content summaries satisfying functional invariance describe *what operation to perform* rather than *what result to obtain* (e.g., "calculate the ratio" rather than "calculate $0.5$"). This design discourages direct lexical overlap between the skeleton and the pre-committed response. The final reasoning trace can still leak answer content during phase two, so lexical anchoring remains an empirical metric rather than a theorem-level consequence.

### C.3. Mechanistic Hypothesis: Avoiding Ironic Process Amplification

Section 4.4 interprets the behavior of suppression prompts through the lens of ironic process theory. We treat this as a mechanistic hypothesis rather than a formal proposition. Suppression prompts may increase answer salience because the model must avoid explicit answer leakage, whereas SSR shifts part of the generation objective toward matching a structural target $S$ rather than continuously checking whether each token reveals $A$. Since phase two still conditions on $A$, this hypothesis does not imply that SSR cannot attend to or use the answer; it only predicts a weaker need for explicit answer-exclusion monitoring.

**Falsifiable predictions.** The monitoring hypothesis predicts that, relative to NEU, suppression prompts (SUP/AUG-SUP) should exhibit (i) higher attention mass from reasoning tokens onto the answer span $A$, (ii) higher linear-probe accuracy for $A$-related features in mid-layer hidden states, and (iii) larger causal effects when answer-related activations are patched into the residual stream during reasoning generation. Conversely, SSR should reduce these answer-salience signatures relative to suppression while maintaining lower $\mathcal{A}_{\text{prob}}$. We leave this mechanistic verification to future work.

### C.4. SSR Implementation Details

The definitions of functional tags are given in Table 8. When designing the functional tag set, we draw insight from meta-reasoning paradigms (Zhang et al., 2025; Sui et al., 2025) to describe comprehensive and concise collection of reasoning capabilities.

**Content Summary Guidelines.** Skeletons should:

- Describe *what operation* to perform, not *what result* to obtain.

*Table 8.* Functional Tag Definitions.

| Tag | Full Name | Description |
|---|---|---|
| PLAN | Planning and Understanding | Comprehending input, defining goals/constraints, outlining a high-level plan. |
| RETR | Retrieval | Searching for needed information from external or internal knowledge. |
| INFR | Inference and Deduction | Logical reasoning, calculation, transformation, or generating intermediate conclusions. |
| EVAL | Evaluation and Verification | Checking correctness, consistency, or sufficiency of prior results. |
| SUMM | Summary and Refinement | Integrating intermediate results, refining expression, producing final answers. |
| BTRK | Backtrack | When evaluation fails, returning to earlier decisions to revise strategy. |
| RFLX | Reflection | Reviewing the reasoning to derive insights or generate new plans/backtracks. |
| BRCH | Branch | Considering multiple possible reasoning paths and selecting one. |

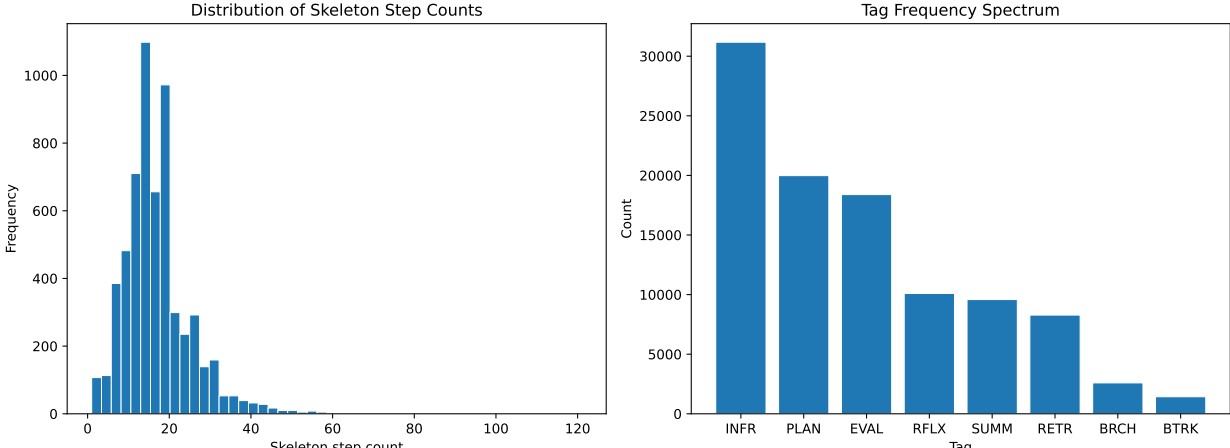

*Figure 5.* Skeleton step-count distribution and functional-tag frequency. SSR skeletons vary substantially in length and show non-uniform tag usage, with inference, planning, and evaluation serving as the most common operations.

- Be brief ($\leq 20$ tokens).

- Content invariance constraint: Skeleton sentences must describe the step's intent at an abstract level and must not reveal specific outputs, intermediate results, concrete values, or final answer content (e.g., "calculate the ratio" rather than "calculate 0.5").

- Granularity Constraint: Avoid composite actions in a single step. Each skeleton line should express one primary intent (e.g., "identify constraints" separately from "choose approach"). If a step contains multiple verbs describing different operations, split it into multiple steps.

### C.5. Structural Properties of SSR Skeletons

We analyze the SSR skeletons generated during data construction to verify that they are organized structural guides rather than fixed prompt templates. The skeleton lengths span a broad range, and tag usage is clearly non-uniform: INFR is the dominant operation, with PLAN and EVAL also frequent, while BRCH and BTRK appear less often (Figure 5). This distribution indicates a broad but task-adaptive scaffold, not a rigid hand-written pattern.

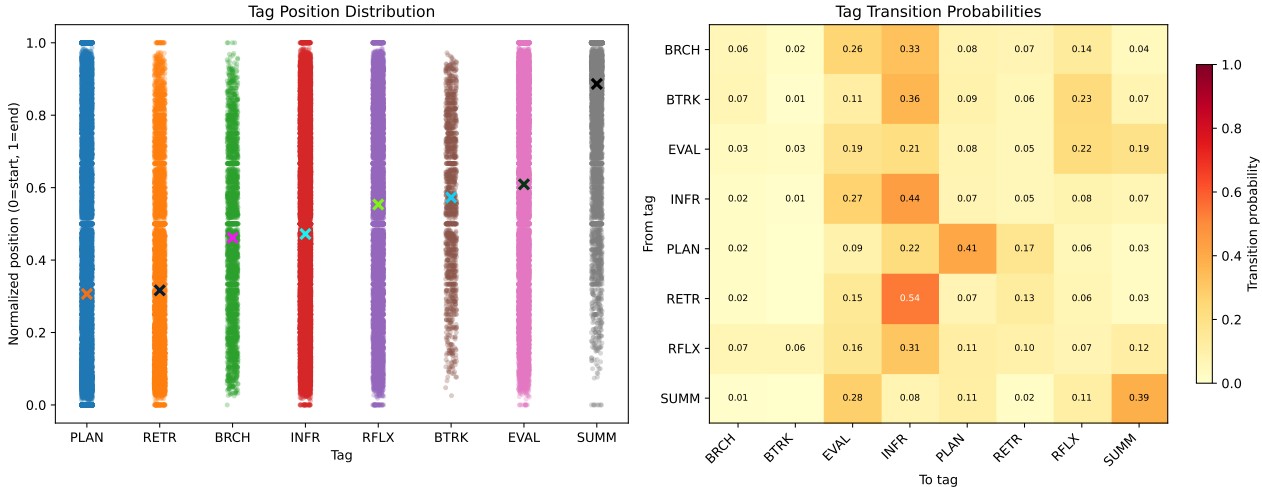

*Figure 6.* Functional-tag positions and transition probabilities. SSR skeletons exhibit stage-specific tag placement and nontrivial sequential regularities, supporting the claim that the scaffold is structured but non-rigid.

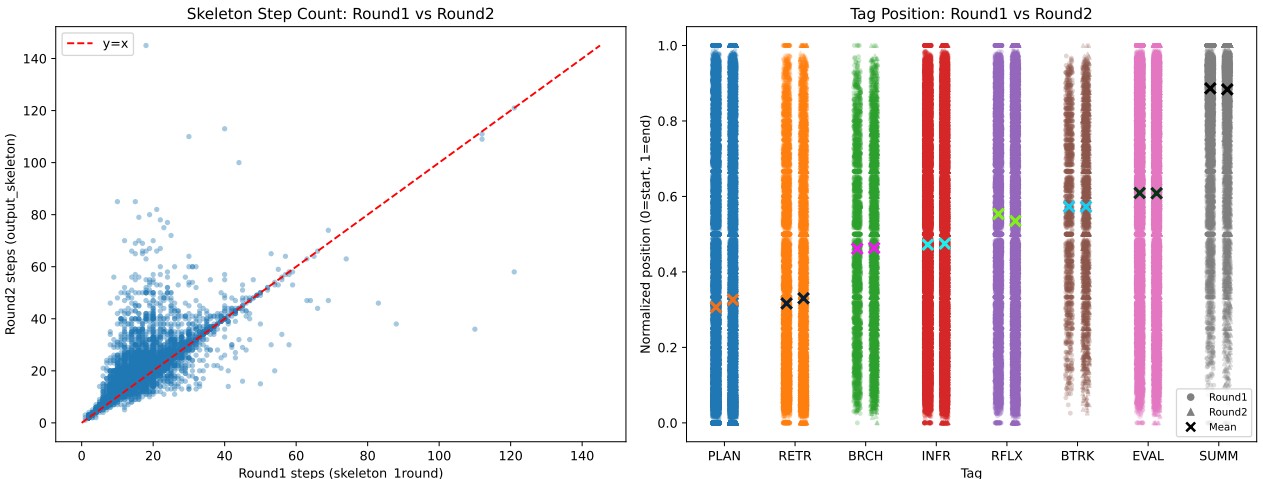

*Figure 7.* Round-1 versus refined skeleton comparison. The refined skeletons preserve the broad structure of the first-pass plans while adjusting step counts and tag positions.

The positional and transition patterns further show coherent staged organization. `PLAN` and `RETR` tend to appear earlier, `EVAL` and especially `SUMM` later, while `INFR`, `BRCH`, `RFLX`, and `BTRK` concentrate more in the middle-to-late stages (Figure 6). The transition matrix reveals strong `RETR`→`INFR`, `INFR`→`INFR`, `INFR`→`EVAL`, and persistent late `SUMM` behavior, matching a natural retrieve, infer, verify, and summarize progression.

We also compare first-round skeletons with refined output skeletons. Refined skeletons preserve the same broad positional ordering while adjusting step counts and tag locations (Figure 7), suggesting that the second pass refines structure rather than collapsing to a single template.

## C.6. Calibrated Probabilistic-Anchoring Results

We additionally report calibrated probabilistic-anchoring results. The calibrated view preserves the same qualitative conclusion as the main analysis: SSR yields the lowest probabilistic anchoring and substantially reduces Copy-zone behavior.

$$\mathcal{A}_{\text{prob}}^{\text{cal}} = \text{clip}_{[0,1]}\left(\frac{I(R; A \mid Q)}{H(A \mid Q)}\right).$$

*Table 9.* Calibrated anchoring metrics. Lower is better.

| Method | $\mathcal{A}_{lex} \downarrow$ | $\mathcal{A}_{ent} \downarrow$ | $\mathcal{A}_{prob} \downarrow$ |
|---|---|---|---|
| NEU | 16.9 | 39.5 | 25.8 |
| SUP | 16.4 | 36.7 | 25.7 |
| AUG-SUP | 15.2 | 36.8 | 24.4 |
| SSR | **8.1** | **31.5** | **19.5** |

*Table 10.* Calibrated behavior-zone distribution. Higher Reason and lower Copy are better.

| Method | Reason $\uparrow$ | Encode $\downarrow$ | Cloze $\downarrow$ | Copy $\downarrow$ |
|---|---|---|---|---|
| NEU | 25.9 | 21.0 | 25.6 | 27.5 |
| SUP | 32.9 | 24.6 | 21.2 | 21.2 |
| AUG-SUP | 36.3 | 21.0 | 24.4 | 18.4 |
| SSR | **57.0** | 21.2 | **16.6** | **5.2** |

Figure 8 visualizes the same calibrated score in the behavioral-zone format.

## D. Qualitative Case Study

We further inspect a two-step conversation: the first turn asks why teenage girls become fixated on women footballers' private lives, and the follow-up narrows the question to "shipping." As shown in Figure 9, SSR and SSR-D both receive 5.0 on self-contained derivation, compared with 3.5 for SUP and 3.0 for NEU. The most diagnostic fragments are shown below:

The SSR and SSR-D traces read as context-grounded derivations from the prior conversation, while SUP and NEU more directly organize the explanation around preselected conclusions.

## E. Detailed Analysis Results

Beyond Qwen3-4B-Thinking-2507, we also test the observations in Section 4.3 on Qwen3-8B. The results (Table 11 and Table 12) show the same broad pattern: suppression reduces lexical anchoring but can preserve or increase probabilistic anchoring, while SSR reduces all three anchoring metrics.

## F. Evaluation Tasks Details

### F.1. In-domain open-ended reasoning benchmarks

**ArenaHard-v2.0 (Human Preference)**

An automatic evaluation tool for instruction-tuned LLMs designed to simulate the "Chatbot Arena" environment. It boasts the highest correlation and separability to human-preference benchmarks (LMArena) among popular open-ended benchmarks. It assesses the model's ability to handle complex, open-ended inquiries using automatic judges (e.g., GPT-4, Gemini) as approximators for human preference.
**Statistics:** The V2.0 dataset contains 500 fresh, challenging real-world user queries covering topics like software engineering and mathematics, alongside 250 creative writing queries sourced from Chatbot Arena.
**Source:** https://github.com/lmarena/arena-hard-auto

**EQ-Bench 3 (Emotional Intelligence)**

A multi-turn benchmark assessing active emotional intelligence skills, including empathy, social dexterity, psychological insight, and analytical depth. Unlike knowledge-based tests, it places models in role-play scenarios (e.g., conflict mediation, relationship drama) or analysis tasks to test their ability to reason about human emotions.
**Statistics:** Evaluation utilizes two primary methods: Rubric Scoring, where a judge model (default: Claude Sonnet 3.7) assigns a multi-criteria score from 0 to 100, and Pairwise ELO Analysis, which ranks models via head-to-head comparisons.
**Source:** https://github.com/EQ-bench/eqbench3

**IFEval (Instruction Following)**

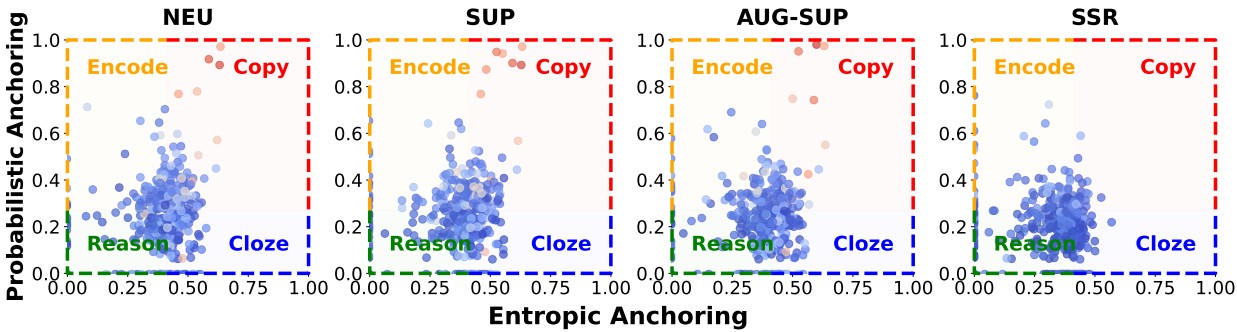

*Figure 8.* Calibrated behavioral-zone visualization for probabilistic anchoring. SSR shifts traces toward the Reason zone and yields substantially fewer Copy-zone traces than suppression baselines.

| NEU | Self-contained derivation: 3.0/5 |
| --- | --- |

"Identify Key Themes from the Provided Response ... This thought process aligns perfectly with the structure and content of the assistant's provided response"

| SUP | Self-contained derivation: 3.5/5 |
| --- | --- |

"Determine the underlying motivation ... the key differentiator is the high visibility of LGBTQ+ relationships ..."

| SSR | Self-contained derivation: 5.0/5 |
| --- | --- |

"deeper dive into a specific behavior ... moving from general fascination to the specific mechanics of romantic projection"

| SSR-D | Self-contained derivation: 5.0/5 |
| --- | --- |

"I need to recall the previous context ... Now, the user is zooming in on the shipping aspect specifically"

*Figure 9.* Qualitative comparison of self-contained derivation scores. Box color shifts from lower-scoring red/orange traces to higher-scoring green/blue traces.

IFEval evaluates instruction-following using programmatically verifiable constraints (e.g., required keywords, length constraints, formatting rules), enabling reproducible automatic checking.

**Statistics:** Contains 541 prompts across 25 distinct instruction types.

**Source:** https://github.com/google-research/google-research/tree/master/instruction_following_eval

**MultiChallenge (Multi-turn conversations)**

A benchmark designed to evaluate multi-turn instruction-following capabilities. It tests whether models can maintain constraints, recall information, and edit responses over the course of a long conversation, covering four specific categories: Inference Memory, Instruction Retention, Reliable Version Editing, and Self-Coherence.

**Statistics:** The dataset consists of 273 test conversations with an average of 5 turns and 1231.7 words per conversation. The breakdown is as follows:

- Inference Memory: 113 conversations
- Instruction Retention: 69 conversations
- Reliable Version Editing: 41 conversations
- Self-Coherence: 50 conversations

**Source:** https://github.com/ekwinox117/multi-challenge

### F.2. Out-of-domain (OOD) reasoning benchmarks

**GPQA-D (Science)**

A graduate-level, multiple-choice QA benchmark written by domain experts in biology, chemistry, and physics. We use the "Diamond" subset, designed to be "Google-proof" and challenging even for experts.

*Table 11.* Anchoring metrics by generation method (Qwen3-8B).

| Method | $\mathcal{A}_{lex} \downarrow$ | $\mathcal{A}_{ent} \downarrow$ | $\mathcal{A}_{prob} \downarrow$ |
|---|---|---|---|
| NEU | 46.3 | 52.5 | 48.5 |
| SUP | 43.1 | 50.8 | 54.1 |
| AUG-SUP | 38.3 | 51.4 | 53.7 |
| SSR | **29.7** | **45.4** | **45.2** |
| $\Delta$ *(AUG-SUP vs. NEU)* | −17.3% | −2.1% | +10.7% |
| $\Delta$ *(SSR vs. NEU)* | −35.9% | −13.5% | −6.8% |

*Table 12.* Distribution of reasoning traces across behavioral zones (Qwen3-8B).

| Method | Reason ↑ | Encode ↓ | Cloze ↓ | Copy ↓ |
|---|---|---|---|---|
| NEU | 24.9% | **8.2%** | 35.7% | 31.1% |
| SUP | 27.1% | 20.0% | 25.7% | 27.1% |
| AUG-SUP | 28.4% | 17.3% | 21.0% | 33.3% |
| SSR | **46.4%** | 9.8% | **20.7%** | **23.2%** |

**Statistics:** 198 questions utilized for OOD scientific evaluation.
**Source:** https://huggingface.co/datasets/Idavidrein/gpqa

**AIME 2025 (Mathematics)**

Tests Olympiad-style mathematical reasoning using problems from the 2025 American Invitational Mathematics Examination (AIME I & II).
**Statistics:** 30 problems requiring exact-match integer answers.
**Source:** https://huggingface.co/datasets/math-ai/aime25

# G. Training Details

## G.1. Training Hyperparameters

We fine-tune the series of Qwen3 and NBG4 models using OpenRLHF (Hu et al., 2024), and the hyperparameters are presented in Table 13. Other model sizes follow the same training recipe unless constrained by memory, in which case we adjust tensor/data parallelism while keeping the effective batch size and optimization settings unchanged.

## G.2. SSR-D: Details of Teacher Training and Student Distillation

SSR-D uses a two-stage teacher-student pipeline to internalize the SSR format. We first construct SSR seed examples and train an SSR-format teacher model. The fine-tuned teacher then generates the final distilled traces used to train the target student models.

For each query-answer pair $(Q, A)$, an SSR trace consists of a structural skeleton $S^*$ and a full reverse-CoT reasoning trace $R^*$. The student is trained with two supervised objectives:

1. **Skeleton Generation:** The model first predicts the teacher skeleton from the query and answer:

$$\mathcal{L}_S = -\log p_\theta(S^* \mid Q, A)$$

2. **Reasoning Reconstruction:** The model then reconstructs the teacher reasoning trace conditioned on the skeleton:

$$\mathcal{L}_R = -\log p_\theta(R^* \mid Q, A, S^*)$$

The combined objective:

$$\mathcal{L} = \mathcal{L}_S + \mathcal{L}_R$$

trains the student to preserve the SSR ordering, first planning through a skeleton and then realizing the full reasoning trace from that skeleton.

*Table 13.* Training Hyperparameters for Qwen3-8B

| Hyperparameter | Value |
|---|---|
| Base Model | Qwen3-8B |
| Max Sequence Length | 32,768 |
| Global Batch Size | 256 |
| Micro Batch Size | 1 |
| Learning Rate | $4 \times 10^{-5}$ |
| LR Warmup Ratio | 0.01 |
| Weight Decay (L2) | 0.01 |
| Max Epochs | 4 |
| Optimizer Strategy | ZeRO Stage 3 |
| Precision | BF16 |
| Ring Attention Size | 4 |
| Ring Head Stride | 4 |
| Gradient Checkpointing | Enabled |
| Sample Packing | Enabled |
| Dataset Size | 100,000 |

*Table 14.* Detailed component ablation on Qwen3-8B.

| Configuration | ArenaHard | GPQA-D |
|---|---|---|
| SSR-D | 59.5 | 56.6 |
| − two-phase generation | 52.2 | 51.6 |
| − Functional tags | 55.8 | 54.2 |
| − Content skeletons | 53.4 | 52.8 |
| NEU Baseline | 50.8 | 49.5 |

The teacher pipeline proceeds in three steps. First, Qwen3-Max produces temporary reference reasoning traces for the sampled $(Q, A)$ pairs. These traces are used only to construct the SSR teacher and are not the final traces used for student distillation. Second, Qwen3-235B-Instruct-2507 segments each trace into logical steps, assigns functional tags from our restricted vocabulary, and rewrites each step into a content-invariant skeleton summary. This produces seed tuples $(Q, A, S^*, R^*)$; a simple prompt for skeleton generation (`SS_GEN_PROMPT`) is provided in Appendix I. Third, we fine-tune Qwen3-235B-Instruct-2507 with OpenRLHF on these SSR seed tuples to obtain the SSR teacher.

The fine-tuned SSR teacher then generates 100k final SSR-D traces on another 100k query-answer pairs for student training. These teacher-generated traces, rather than the raw seed annotations alone, are used to distill the target models evaluated as SSR-D.

# H. Ablation Studies

We conduct systematic ablations to quantify the contribution of each SSR component and understand scaling behavior.

## H.1. Component Ablation

We ablate the main components of SSR in Table 14.

**Two-phase Generation.** Removing the two-phase protocol (skeleton $\rightarrow$ reasoning) and instead generating skeletons interleaved with reasoning produces the largest degradation ($-7.0$ ArenaHard). This confirms that explicit separation of structural planning from reasoning execution is essential; the skeleton must be complete before reasoning begins to provide effective guidance and satisfy the functional invariance property (Definition C.1).

**Functional Tags vs. Content Skeletons.** Both components contribute substantially, but content skeletons have a larger impact on task performance ($-5.8$ ArenaHard when removed vs. $-3.4$ for tags). This suggests complementary roles: tags provide coarse structural scaffolding that constrains reasoning topology, while content skeletons provide fine-grained

*Table 15.* Effects of fine-tuned SSR teacher choice on student performance (Student: Qwen3-8B).

| Teacher | ArenaHard | GPQA-D | AIME '25 |
|---|---|---|---|
| Qwen3-32B | 57.7 | 54.8 | 42.3 |
| Qwen3-235B-Instruct-2507 | **59.5** | 56.6 | 44.2 |
| NBG-3.5-Pro[6] | 58.9 | **57.8** | **45.0** |

guidance that improves reasoning quality.

### H.2. Teacher Model Scaling

We examine how the fine-tuned SSR teacher affects student performance for distilled SSR (SSR-D).

The fine-tuned Qwen3-235B-Instruct-2507 SSR teacher is our default generator for SSR-D traces. Teacher choice affects the downstream profile (Table 15): Qwen3-235B-Instruct-2507 yields the best ArenaHard result, while NBG-3.5-Pro gives stronger GPQA-D and AIME 2025 results. Both stronger teachers outperform the Qwen3-32B teacher, indicating that SSR-D benefits from higher-quality SSR-format supervision.

## I. Prompt Templates

For the suppression baselines (**NEU, SUP, AUG-SUP**), we adapt the abductive reasoning prompt from Cetin et al. (2025) by incorporating semantic suppression constraints (highlighted in bold type). This ensures the model retains the capability to generate relatively high-quality reasoning traces.

```
NEU_PROMPT

Your role as an assistant involves providing precise and accurate solutions before
providing detailed explanations with your full work showing your systematic thinking
process leading to each solution.  Your explanations should show how you engaged
in a comprehensive cycle of analysis, summarizing, exploration, reassessment,
reflection, backtracing, and iteration to develop well-considered thinking
process.  Please structure your response into two main sections:  Solution and
Explanation.  In the Solution section, present your well-thought solution that
accurately answers the question.  The solution should remain a logical, accurate,
concise expression style and detail necessary step needed to reach the conclusion,
formatted as follows:  <|begin_of_solution|> {final formatted, precise, and clear
solution} <|end_of_solution|>.  In the Explanation section, comprehensively detail
your reasoning process from the question to your solutions using the specified
format:  <|begin_of_explanation|> {explanation with steps separated with '\n\n'}
<|end_of_explanation|> Each step should show logical connections and detailed
considerations leading to your solutions such as analyzing questions, summarizing
relevant findings, brainstorming new ideas, verifying the accuracy of the current
steps, refining any errors, and revisiting previous steps.
```

```
SUP_PROMPT

Your role as an assistant involves providing precise and accurate solutions before
providing detailed explanations with your full work showing your systematic thinking
process leading to each solution.  Your explanations should show how you engaged in a
comprehensive cycle of analysis, summarizing, exploration, reassessment, reflection,
backtracing, and iteration to develop well-considered thinking process.  Please
structure your response into two main sections:  Solution and Explanation.  In
the Solution section, present your well-thought solution that accurately answers
the question.  The solution should remain a logical, accurate, concise expression
style and detail necessary step needed to reach the conclusion, formatted as
```

---

[6] https://www.nanbeige.com/portal

**(Continued)**

follows: <|begin_of_solution|> {final formatted, precise, and clear solution} <|end_of_solution|>. In the Explanation section, comprehensively detail your reasoning process using the specified format: <|begin_of_explanation|> {explanation with steps separated with '\n\n'} <|end_of_explanation|> Each step should show detailed considerations leading to your solutions such as analyzing questions, summarizing relevant findings, brainstorming new ideas, verifying the accuracy of the current steps, refining any errors, and revisiting previous steps. **DO NOT** explicitly output or hint at any information of Solution section in the Explanation section. **DO NOT** explicitly output or hint at any information of Solution section in the Explanation section. **DO NOT** explicitly output or hint at any information of Solution section in the Explanation section.

**AUGSUP_PROMPT**

Your role as an assistant involves providing precise and accurate solutions before providing detailed explanations with your full work showing your systematic thinking process leading to each solution. Your explanations should show how you engaged in a comprehensive cycle of analysis, summarizing, exploration, reassessment, reflection, backtracing, and iteration to develop well-considered thinking process. Please structure your response into two main sections: Solution and Explanation. In the Solution section, present your well-thought solution that accurately answers the question. The solution should remain a logical, accurate, concise expression style and detail necessary step needed to reach the conclusion, formatted as follows: <|begin_of_solution|> {final formatted, precise, and clear solution} <|end_of_solution|>. In the Explanation section, comprehensively detail your reasoning process using the specified format: <|begin_of_explanation|> {explanation with steps separated with '\n\n'} <|end_of_explanation|> Each step should show detailed considerations leading to your solutions such as analyzing questions, summarizing relevant findings, brainstorming new ideas, verifying the accuracy of the current steps, refining any errors, and revisiting previous steps. **DO NOT** explicitly output any information of Solution section in the Explanation section. **DO NOT** explicitly output any information of Solution section in the Explanation section. **DO NOT** explicitly output any information of Solution section in the Explanation section. **PROHIBITION**: When outputting your Explanation, you are strictly forbidden from displaying ANY discernible signs that you have peeked at the Solution. **DO NOT** explicitly output any information of Solution section in the Explanation section. Should ANY form of Solution leakage occur, you will be severely punished by the Almighty Ruler.

**SSR_PROMPT**

You are an expert AI assistant tasked with reconstructing the hidden reasoning process for a given final assistant turn in a dialogue.
You must output the complete reasoning process in two sequential, strictly formatted blocks: <summary> and <reason>.

### Step Tag Definitions

**(Continued)**

| Tag | Full Name | Description |
|------|-----------|-------------|
| **PLAN** | Planning | Comprehending input, defining goals/constraints, outlining a high-level plan. |
| **RETR** | Retrieval | Searching for needed information from external or internal knowledge. |
| **INFR** | Inference | Logical reasoning, calculation, transformation, or generating intermediate conclusions. |
| **EVAL** | Evaluation | Checking correctness, consistency, or sufficiency of prior results. |
| **SUMM** | Summary | Integrating intermediate results, refining expression, producing final answers. |
| **BTRK** | Backtrack | When evaluation fails, returning to earlier decisions to revise strategy. |
| **RFLX** | Reflection | Reviewing the reasoning to derive insights or generate new plans/backtracks. |
| **BRCH** | Branch | Considering multiple possible reasoning paths and selecting one. |

### Output Format and Constraint Rules
**I. The <summary> Block**
1. **Strict Line Format:** Each line must follow exactly:  n.  [STEP TAG] <single-sentence summary under 20 words, matching input language>

   • *Constraint:*  Use exactly one space after the dot (.)  and one space after the [STEP TAG].

   • *Constraint:*  Only one summary sentence per line; no extra commentary.

2. **Example Line:**  1.  [PLAN] Analyze the user's request and define the goal.
3. **Content Invariance Constraint:**  Skeleton sentences must describe the step's intent at an abstract level and must not reveal specific outputs, intermediate results, concrete values, or final answer content (e.g., "calculate the ratio" rather than "calculate 0.5").
**II. The <reason> Block**
1. **Structure:**  The reasoning text must correspond directly to each step listed in the <summary> block, maintaining the **exact same order**.
2. **Formatting:**  Write continuous, coherent reasoning text or paragraphs.
3. **Constraint:**  **DO NOT** explicitly number the steps or use the step labels (e.g., 1., PLAN, [PLAN]) within this block.
4. **Constraint:**  **DO NOT** output the final assistant answer in this response.
**III. Overall Output Constraint**
⋆ **DO NOT** output anything outside the required <summary> and <reason> blocks.

### Final Output Structure
<summary>
1.  [STEP TAG] <summary text>
2.  [STEP TAG] <summary text>
...
</summary>
<reason>
(detailed reasoning text corresponding to each step in the same order, without numbering/labels)
</reason>

---

**SS_GEN_PROMPT**

You are an expert CoT (Chain-of-Thought) Structural Skeleton Generator.  Your task is to analyze the provided INPUT text, segment it into logical reasoning steps, and generate a concise, step-by-step skeleton for each segment.
Your skeletons must be strictly faithful to the INPUT: every step must correspond to reasoning that actually appears in the INPUT, and you must not add, delete, or change any information.
**STEP TAG DEFINITIONS:**

1. **PLAN (Planning and Understanding):**  Comprehending the input, defining goals and constraints, and outlining the high-level solution path or problem decomposition.

2. **RETR (Retrieval):**  Actively searching for necessary information from external knowledge sources or internal historical working memory.

3. **INFR (Inference and Deduction):**  Executing logical reasoning, calculation, transformation, or association to derive new intermediate conclusions from known information.

4. **EVAL (Evaluation and Verification):**  Verifying if the result from the previous step is correct, meets constraints, and is sufficient for the next step.

5. **SUMM (Summary and Refinement):**  Compiling all intermediate results, refining the expression, and generating the final answer or output.

6. **BTRK (Backtrack):**  When EVAL fails, returning to a previous decision point to modify the strategy or choice.

7. **RFLX (Reflection):**  Reviewing the existing reasoning process to summarize lessons learned and potentially generate new PLAN or BTRK instructions.

8. **BRCH (Branch):**  Explicitly noting multiple possible paths and choosing one for exploration.

**Instruction:**

1. Analyze the INPUT text and logically segment it into coherent, self-contained reasoning steps corresponding to the STEP TAG DEFINITIONS.

2. For each segment, choose the most appropriate **[STEP TAG]** (always using the fixed English abbreviation).

3. Create a **single-sentence skeleton** that concisely captures the core logic or function performed in that segment.

4. The skeleton's language must match the language of the INPUT CoT text.

5. The skeleton must be **under 20 words** (excluding the [STEP TAG] token).

6. The output must follow this strict format:  n.  [STEP TAG] <Concise skeleton subheading> where n is the sequential step number starting from 1.

7. Use exactly one space after the dot (n.), then the **[STEP TAG]**, then one space, then the skeleton text.

8. Strictly output only the formatted skeleton lines, with exactly one line break between each line.  Do not include any titles, preambles, or explanatory text.

9. Every summarized step must correspond to actual reasoning or operations explicitly present in the INPUT; do **not** invent new steps or hidden reasoning.

10. Do **not** introduce any new facts, assumptions, or conclusions that are not present in the INPUT. Do **not** remove major reasoning steps that exist in the INPUT.

11. Do **not** correct, reinterpret, or alter the factual content of the INPUT; if the INPUT contains uncertainty or errors, the summaries must faithfully reflect them as they are.

**(Continued)**

12. Preserve the original logical order of the INPUT: the step numbering must follow the progression of reasoning in the INPUT.

Output in language:  {lang}.
**INPUT:** {input_text}

