# OpenReview forum: "Measuring and Mitigating Post-Hoc Rationalization in Reverse Chain-of-Thought Generation"
_ICML.cc/2026/Conference — ICML 2026 regular_

### Official Review · Reviewer_AXJQ · 2026-03-09

**Soundness:** 3
**Presentation:** 3
**Significance:** 3
**Originality:** 3
**Overall Recommendation:** 4
**Confidence:** 3

**Summary:**

The authors tackle a big problem in Reverse Chain of Thought generation. When large language models already know the final answer they tend to cheat by working backward and just rationalizing the result instead of truly reasoning. To study this the paper introduces three new ways to measure how much the answer anchors the reasoning process covering lexical entropic and probabilistic levels. The team discovers that simply telling the model to ignore the answer fails badly because of an ironic monitoring effect where the model thinks about the answer even more. To solve this the authors propose Structural Skeleton guided Reasoning. This method first builds an abstract step by step plan without the exact answer and then uses that plan to guide the full reasoning. They also created a distilled version to train models directly on these good reasoning paths.

**Compliance With Llm Reviewing Policy:**

Affirmed.

**Final Justification:**

The three-level anchoring hierarchy is a creative way to measure post-hoc rationalization, and using Ironic Process Theory to explain why semantic suppression backfires is clever and well-grounded.

The rebuttal covered my concerns effectively — the compute fairness issue is resolved and the LLM-as-judge experiment provides a workable proxy for black-box settings. The main lingering issue is relying on a single teacher model for data construction. Skeleton-guided reasoning isn't entirely new, but applying it as an answer-invariant anti-anchoring mechanism in reverse-CoT is a meaningful twist. Good contribution overall.

**Key Questions For Authors:**

Q1: How much extra time does it take to generate the structural skeleton compared to just running the neutral baseline?
Q2: Have you thought about ways to estimate the entropic and probabilistic anchoring for black box models where we cannot see the token probabilities?
Q3: Does the skeleton generator ever get completely confused if the user prompt is intentionally tricky or misleading?

**Limitations:**

yes

**Strengths And Weaknesses:**

Soundness
Strength: Bringing in Ironic Process Theory from psychology to explain why semantic suppression fails is brilliant and makes total sense  . The mathematical foundation for the entropic and probabilistic metrics is also very robust.
Weakness: Your entropic and probabilistic metrics require access to the internal token probabilities of the models. This means we cannot easily use these metrics on closed commercial models. Also comparing your two phase method against a single phase baseline might give your method an unfair advantage in compute time.

Presentation
Strength: The writing is clear and engaging. The visual figures do a fantastic job of explaining the core problem and your proposed solution.
Weakness: Some of the results tables are incredibly dense with numbers. It might be much easier for readers to digest if you swapped a few of these dense tables for simple visual bar charts.

Significance
Strength: Finding ways to generate high quality synthetic reasoning data is one of the biggest challenges in AI right now. By fixing the post hoc rationalization issue your work provides a massive step forward for training better reasoning models.
Weakness: Forcing the model to generate a skeleton first adds extra generation time which could limit how people use it in fast paced pipelines.

Originality
Strength: Moving past simple word overlap to measure the actual information flow and entropy is a super creative and original approach.

---

> ### Author Rebuttal · Authors · 2026-03-31
>
> We thank the reviewer for the encouraging assessment of the psychological framing, the mathematical grounding of the metrics, and the overall significance of the work.
>
> **On extra time from skeleton generation / compute fairness.** The additional overhead is small in practice. The structural skeleton typically accounts for only **1–5% of the final CoT length**, so its generation cost is negligible relative to the full reasoning trace under a roughly token-proportional decoding model. Moreover, this cost is incurred only during **data construction**. The deployed student model performs inference in a **single forward pass**, so there is no extra runtime overhead at deployment. We agree that it is important to state this more clearly, and we will add an explicit discussion of compute-time fairness and construction-vs-inference cost in the final version.
>
> **On estimating anchoring metrics for black-box models.** We agree that the current entropic and probabilistic metrics are most naturally defined for open-weight or otherwise probability-accessible models, which limits direct applicability to closed commercial APIs. To partially address this, we conducted the **500-sample LLM-as-judge study** (see Reviewer Ub1N for details). Importantly, that evaluation operates purely at the **input/output level**: in the self-contained-derivation pass, the evaluator sees only the query and reasoning trace; in the post-hoc-dependence pass, it additionally sees the pre-committed answer and judges how answer-driven the trace appears. Across both **claude-opus-4-6** and **gpt-5.4**, SSR-D is ranked best and SSR outperforms SUP and NEU, indicating that the reasoning-quality differences captured by our anchoring framework are also visible in output-level assessments that do **not** require token probabilities. We therefore view output-level LLM-judge evaluation as a practical proxy when internal probabilities are unavailable, while acknowledging that it is not a drop-in replacement for the internal metrics. We will discuss this direction more explicitly in the final version.
>
> **On skeleton-generator stability under tricky or misleading prompts.** We do not yet have a dedicated stress test for adversarial or misleading inputs, and we agree this is a limitation. Because the skeleton capacity is limited, its failure mode in such cases would more likely be **structural incoherence or poor planning** rather than anchoring-related leakage. Characterizing these boundary conditions is an important direction for future work, and we will add this limitation to the final version.
>
> **On presentation.** Thanks for pointing out the presentation issue. We will improve the presentation of the numerical results in the tables and provide a more readable visual version in the final paper.

---

> > ### Author Rebuttal · Reviewer_AXJQ · 2026-04-03
> >
> > Thank you for the rebuttal. My concerns have been adequately addressed. I maintain my score of 4.

---

> > > ### Author Response · Authors · 2026-04-08
> > >
> > > Thank you for your thoughtful review. We are glad that our rebuttal addressed your concerns.

---

### Official Review · Reviewer_Ub1N · 2026-03-10

**Soundness:** 3
**Presentation:** 3
**Significance:** 3
**Originality:** 3
**Overall Recommendation:** 4
**Confidence:** 3

**Summary:**

This paper examines the post-hoc rationalization in Reverse Chain-of-Thought Generation (RCG) process and proposes a three-level measurement hierarchy to measure it. Given that, the paper proposes Structural Skeleton-guided Reasoning (SSR) to alleviate anchoring to answer.

**Compliance With Llm Reviewing Policy:**

Affirmed.

**Final Justification:**

This paper is technically solid and well presented, the author's rebuttal addresses most of my concerns.

**Key Questions For Authors:**

1. Can you show more empirical evidence on the validity of proposed entropic/probabilitic anchoring?

**Limitations:**

yes

**Strengths And Weaknesses:**

## Strengths
1. The paper is overall well-written and easy to read.
2. The empirical results show strong improvement on downstream tasks.

## Weaknesses
1. Although the proposed Entropic anchoring and probablitic anchoring are well motivated, the lack of empirical evidence poses doubts on the validity of the two metrics, which are central in the evaluation and motivate the design of SSR method.
2. The idea of constructing structural reasoning traces (SSR), is not very novel and applied in many general reasoning subfields.

---

> ### Author Rebuttal · Authors · 2026-03-31
>
> We thank the reviewer for the helpful comments and for recognizing the empirical strength of the downstream results.
>
> **On empirical validation of entropic and probabilistic anchoring.** We agree that stronger external validation is valuable. To this end, we conducted a lightweight **500-sample LLM-as-judge study** using **claude-opus-4-6** and **gpt-5.4**. The evaluation used two separate passes: (i) **self-contained derivation**, where the evaluator saw only the query and reasoning trace, and (ii) **post-hoc dependence**, where the evaluator additionally saw the pre-committed answer and judged how answer-driven the trace appeared.
>
> The quantitative pattern is highly consistent across both judges, as shown below.
>
> **claude-opus-4-6**
>
> | Method | Self-contained derivation ↑ (mean) | Variance | Post-hoc dependence ↓ (mean) | Variance |
> |---|---:|---:|---:|---:|
> | **SSR-D** | **4.7442** | 0.2632 | **2.6977** | 0.6840 |
> | **SSR** | **4.6146** | 0.4920 | **3.4479** | 0.7762 |
> | SUP | 4.4388 | 0.9292 | 3.5918 | 1.0276 |
> | NEU | 4.3750 | 0.8895 | 3.5625 | 1.2171 |
>
> **gpt-5.4**
>
> | Method | Self-contained derivation ↑ (mean) | Variance | Post-hoc dependence ↓ (mean) | Variance |
> |---|---:|---:|---:|---:|
> | **SSR-D** | **4.4100** | 0.3656 | **3.0400** | 1.3115 |
> | **SSR** | **4.1134** | 0.7683 | **4.1546** | 1.4446 |
> | SUP | 3.6900 | 1.5292 | 4.3800 | 1.2077 |
> | NEU | 3.5800 | 1.4582 | 4.4800 | 0.9188 |
>
> Thus, under both evaluators, **SSR-D is ranked best on both dimensions**, and **SSR consistently outperforms SUP and NEU**. We view this as convergent external evidence that lower anchoring aligns with independently judged improvements in reasoning quality and reductions in post-hoc rationalization.
>
> We also have two complementary forms of qualitative evidence. First, our structural analysis shows that SSR-generated reverse CoTs have a broad length distribution, a nontrivial tag-frequency profile, clear positional specialization across tags, and meaningful transition regularities rather than a fixed template. Concretely, **INFR** dominates overall usage; **PLAN/RETR** are earlier-stage operations; **EVAL/SUMM** shift later; and transitions such as **RETR→INFR**, **INFR→INFR**, **INFR→EVAL**, and persistent late **SUMM** behavior indicate coherent staged organization. Second, the case study described above provides a concrete example of what this difference looks like in practice. In brief, SSR/SSR-D contain context-grounded fragments such as *“recall the previous context”* or *“moving from general fascination to the specific mechanics ...”*, while SUP/NEU contain more answer-conditioned fragments such as *“Determine the underlying motivation ...”* or *“Identify Key Themes from the Provided Response ...”* This makes the distinction human-readable in addition to being visible in the metric tables. We will add this analysis to the final version. These results do not “prove” the metrics are perfect, but they provide additional empirical support for their validity.
>
> **On novelty relative to prior structural reasoning work.** We agree that skeleton-guided and plan-first reasoning methods exist. Our claim is not that “structure” itself is new, but that **SSR is distinct in the reverse-CoT setting because the scaffold is answer-invariant**. In our setting, the model has access to a visible target answer, and the main failure mode is post-hoc rationalization around that answer. SSR is designed precisely to break this anchoring cycle by separating planning structure from answer content. In other words, the novelty is not simply using a skeleton, but using an **answer-invariant scaffold as an anti-anchoring mechanism in reverse-CoT generation**. We will revise the related-work discussion to make this distinction more explicit.

---

> > ### Author Rebuttal · Reviewer_Ub1N · 2026-04-02
> >
> > Thank you for the rebuttal. I'll maintain my original score.

---

> > > ### Author Response · Authors · 2026-04-08
> > >
> > > Thank you for your thoughtful review. We are glad that our rebuttal addressed your concerns.

---

### Official Review · Reviewer_2ut7 · 2026-03-11

**Soundness:** 3
**Presentation:** 3
**Significance:** 3
**Originality:** 3
**Overall Recommendation:** 4
**Confidence:** 4

**Summary:**

This paper studies the problem of post-hoc rationalization in Reverse Chain-of-Thought Generation (RCG) for large language models. The authors propose a three-level hierarchy (i.e., lexical, entropic, and probabilistic anchoring) to quantify the anchoring effect introduced by pre-committed answers, and show that conventional semantic suppression strategies can paradoxically worsen the problem due to an ironic process. To mitigate this issue, the paper introduces SSR and SSR-D (distilled variant), which separate reasoning structure from answer semantics.

**Compliance With Llm Reviewing Policy:**

Affirmed.

**Final Justification:**

My initial recommendation towards this paper was a weak acceptance;

During the rebuttal phase, the authors' additionally provided content addressed most of my concerns;

I adjust my confidence score from 3 to 4 and finally rate this paper as a weak acceptance to clear acceptance.

**Key Questions For Authors:**

(i) The data construction pipeline relies on Qwen3-Max for both response generation and self-evaluation. Could the authors discuss whether this design might introduce model-specific biases in the generated reasoning traces?

(ii) Could the authors provide more qualitative examples to illustrate how SSR/SSR-D mitigates rationalization compared with baselines?

(iii) SSR/SSR-D introduce additional stages such as skeleton extraction and two-phase generation. What is the resulting computational overhead (e.g., inference latency or training cost)?

**Limitations:**

yes

**Strengths And Weaknesses:**

strengths:

(i) Introduces a clear multi-level anchoring framework (lexical, entropic, and probabilistic), which goes beyond surface-level lexical overlap to capture latent dependence and generation dynamics.

(ii) The “behavioral zones” construction (Reason/Cloze/Encode/Copy) provides an interpretable conceptual and empirical framework for understanding model behavior within the anchoring metric space.

(iii) Proposes SSR and SSR-D to decouple reasoning structure from semantic content as an approach to mitigate anchoring. The method is supported by a theoretical framing (e.g., capacity bound and monitoring bypass) that connects the mitigation mechanism to the proposed anchoring metrics.

weaknesses:

(i) The data construction pipeline relies heavily on a single model (Qwen3-Max) for both response generation and self-evaluation, which may introduce model-specific bias and limit the diversity of reasoning patterns in the dataset.

(ii) The paper would benefit from more qualitative examples and case studies. In particular, it is not always clear what the constructed data look like or how SSR qualitatively mitigates rationalization compared with baseline methods.

(iii) SSR/SSR-D introduce additional generation and training stages (e.g., skeleton extraction and two-phase generation). However, the paper does not discuss the resulting impact on computational efficiency and latency.

---

> ### Author Rebuttal · Authors · 2026-03-31
>
> We thank the reviewer for the positive assessment of the anchoring framework and behavioral-zone analysis.
>
> **On possible single-model bias from Qwen3-Max.** We agree this remains a limitation of the current study. Two points partially mitigate it. First, the student models trained on our data are distinct from the teacher: the paper includes results for **NBG4-3B-Base**, a model from a different family than Qwen3, which also benefits from SSR-D training data, suggesting the gains are not architecture-specific. Second, the anchoring problem is a property of the **reverse-CoT generation process itself**: any model asked to construct a justification for a visible pre-committed answer faces the same answer-conditioning and ironic-process dynamic regardless of which teacher produced the answer. That said, we agree that using a more diverse set of teacher models would strengthen the generalization claim, and we will make this limitation and future direction explicit in the final version.
>
> **On qualitative examples and case studies.** We agree and have prepared a qualitative case study using the same two-pass evaluation logic as our metric-validation analysis. In a two-step conversation—first a general question about why teenage girls become fixated on women’s footballers’ private lives, then a narrower follow-up about “shipping”—SSR and SSR-D both received **5.0** for self-contained derivation, whereas SUP received **3.5** and NEU **3.0**. To keep the example concise, the most diagnostic fragments are:
>
> - **SSR:** *“deeper dive into a specific behavior ... moving from general fascination to the specific mechanics of romantic projection”*
> - **SSR-D:** *“I need to recall the previous context ... Now, the user is zooming in on the shipping aspect specifically”*
> - **SUP:** *“Determine the underlying motivation ... the key differentiator is the high visibility of LGBTQ+ relationships ...”*
> - **NEU:** *“Identify Key Themes from the Provided Response ... This thought process aligns perfectly with the structure and content of the assistant’s provided response”*
>
> The contrast is that **SSR / SSR-D read as context-grounded derivation from the prior conversation**, whereas **SUP / NEU read increasingly like answer-conditioned organization around preselected conclusions**. In particular, NEU most directly signals reverse-engineering from the provided answer, while SSR-D most clearly retrieves prior conversational context and refines the task from that context. We will include this case study in the final version.
>
> We also performed a population-level structural analysis of SSR-generated reverse CoTs. The analysis shows that skeleton lengths span a broad range rather than collapsing to a fixed template, and that tag usage is clearly non-uniform: **INFR** is the dominant tag, with **PLAN** and **EVAL** also frequent, while **BRCH** and **BTRK** appear less often. Positional patterns are likewise structured rather than arbitrary: **PLAN** and **RETR** tend to appear earlier, **EVAL** and especially **SUMM** later, while **INFR / BRCH / RFLX / BTRK** concentrate more in the middle-to-late stages. The transition analysis further reveals nontrivial sequential regularities, such as strong **RETR→INFR**, **INFR→INFR**, **INFR→EVAL**, and persistent late **SUMM** behavior. Together, these observations suggest that SSR-RCoTs follow an organized but non-rigid scaffold rather than a trivial prompt artifact or fixed hand-written template. We will add this structural analysis to the final version.
>
> **On computational overhead / latency.** The additional overhead of SSR is minimal in practice: the structural skeleton typically accounts for only **1–5% of the final CoT length**, so its generation cost is negligible relative to the full reasoning trace under a roughly token-proportional decoding model. Importantly, this cost is incurred only during **training data construction**. At deployment time, the distilled student model performs inference in a **single forward pass**, so the method does not impose an extra runtime burden on end users. We will add an explicit efficiency discussion to the final version.

---

> > ### Author Rebuttal · Reviewer_2ut7 · 2026-04-03
> >
> > I think the concerns regarding Q(i) still remain. But I overall lean to weak acceptance of the paper.

---

> > > ### Author Response · Authors · 2026-04-08
> > >
> > > Thank you for your thoughtful follow-up and for continuing to engage with the paper. We agree that relying on Qwen3-Max as the sole model for both response generation and self-evaluation leaves room for model-specific bias, and we also agree that using more diverse models is an important way to strengthen the generalization claim.
> > >
> > > To further address this point, we additionally repeated the same experiment setting used in our rebuttal to Reviewer AHUG (i.e., 16k subset), but with Kimi2.5 as the model for both response generation and self-evaluation. In that setting, we obtain:
> > >
> > > | Method | Score |
> > > |---|---:|
> > > | NEU | 35.4 |
> > > | SUP | 36.9 |
> > > | AUG-SUP | 37.5 |
> > > | **SSR (Ours)** | **39.6** |
> > > | **SSR-D (Ours)** | **44.0** |
> > >
> > > The pattern is consistent with our main results: SSR improves over the answer-conditioned baselines, and SSR-D remains the strongest variant. We believe it meaningfully strengthens the evidence that the benefit of structure-guided reverse CoT generation is not specific to Qwen3-Max alone. We will add this cross-model result and clarify this point in the final version.

---

### Official Review · Reviewer_AHUG · 2026-03-16

**Soundness:** 4
**Presentation:** 3
**Significance:** 3
**Originality:** 4
**Overall Recommendation:** 5
**Confidence:** 4

**Summary:**

The authors propose SSR, a two-phrase method that generates a skeleton structure to solve the problem and then on the second stage, uses this skeleton to generate the full reasoning trace. To test the quality of CoT, the authors use (1) lexical anchoring: overlap between the CoT & answer, (2) entropic anchoring: the geometric mean between global uniformity and local non-uniformity (which needs to balance between exploration and coherence), and (3) probabilistic anchoring: the average reduction in uncertainty of predicting the answer given the intermediate reasoning. The authors derive many interesting observations in Section 4.3, and demonstrate that such analyses could lead to better synthetic data generation in Section 5.

**Compliance With Llm Reviewing Policy:**

Affirmed.

**Key Questions For Authors:**

* From the paper "Demystifying Long Chain-of-Thought Reasoning in LLMs", there was an experiment in Table 1 (from the Yeo et al., paper)  that showed that constructred CoTs (which is equivalent to using skeletons in this work) achieve lower performances than emergent CoTs (which is equivalent to neutral baseline in this work). Given that this work gained a lot of attention from the community, can you add a section explaining why your experiments led to contrasting conclusions? I am really curious about your opinions here.

**Limitations:**

yes

**Strengths And Weaknesses:**

* The experiments are really comprehensive and well-organized: I really liked that part that the analyses in Section 4 could translate to better performance in Section 5.
* In section 5.1, it is surprising that training on both skeleton generation and reasoning construction (SSR-D) leads to the best performances compared to baselines.

As for weaknesses,
* In Table3, a commonly used baseline for synthetic data generation is to generate CoTs without providing the answer and then filter out the rationales where the prediction is different with the answer. This might be more applicable to math training datasets than LMArena, but you could try this by using a reward model for filtering. I think this is an essential baseline to compare "filtering-based approaches" vs "reverse CoT approaches" for synthetic data generation.

---

> ### Author Rebuttal · Authors · 2026-03-31
>
> We thank the reviewer for the positive assessment of the paper, especially for noting the strong connection between the anchoring analysis in Section 4 and the downstream gains in Section 5, and for highlighting the effectiveness of SSR-D.
>
> **On the missing filtering-based baseline.** We agree this comparison is important. We note, however, that our main setting differs from standard rollout-filtering setups because the reverse-CoT target is an **enhanced response** constructed by iterative self-refinement (Appendix B), rather than a single sampled answer paired with its native CoT. The outline of the iterative self-refinement is given below:
>
> 1. **Candidate generation**
>    Generate \(N\) independent response rollouts for the same query to encourage diversity.
>
> 2. **Candidate evaluation**
>    Self-evaluate each rollout using a scalar score based on **accuracy, coherence, and completeness**.
>
> 3. **Candidate aggregation**
>    Construct a new set of \(K\) candidates. For each slot, randomly sample \(M\) evaluated candidates and synthesize an improved response by combining the strongest components.
>
> 4. **Improvement loop**
>    Feed the synthesized candidates back into evaluation and aggregation for \(T\) refinement loops, then return the highest-scoring final answer.
>
> The final target answer is the product of multiple rounds of generation, evaluation, and aggregation, while its intermediate reasoning remains implicit. This makes the setting substantially harder: even the best-of-\(n\) native rollout CoT is not directly aligned with the final enhanced response that must later be justified.
>
> To assess this concern empirically, we additionally ran a filtering-style comparison on a 16k subset using Qwen3-Max-Preview rollouts with PPL-based selection. The results are shown below.
>
> | Method | Score |
> |---|---:|
> | Qwen3-Max-Preview | 45.2 |
> | Qwen3-Max-Preview (best among 3 rollouts) | 46.7 |
> | NEU | 44.0 |
> | SUP | 45.5 |
> | AUG-SUP | 45.8 |
> | **SSR (Ours)** | **48.2** |
> | **SSR-D (Ours)** | **55.1** |
>
> These results suggest that, under the enhanced-response setting, SSR-generated reverse CoTs are not only competitive with filtering-based CoTs, but can surpass them. We will add this comparison and the setting clarification to the final version, and will open-source the 16k subset upon acceptance.
>
> **On the discrepancy with Yeo et al.** We agree that Yeo et al. deserves discussion, but we do not believe their “constructed CoT” is equivalent to our SSR, nor their “emergent CoT” equivalent to our NEU baseline. In Yeo et al., Table 1 compares two sources of long-CoT training data in a standard forward-reasoning setup: “constructed” trajectories are built by a multi-step Action Prompting framework with primitive actions such as *clarify, decompose, solution_step, reflection,* and *answer*, whereas “emergent” trajectories are distilled from an existing long-CoT teacher. By contrast, our paper studies **reverse CoT generation with a visible pre-committed answer**, where the core failure mode is **post-hoc rationalization induced by answer anchoring**. NEU is simply the default answer-conditioned reverse-generation baseline rather than an “emergent long-CoT” teacher condition. SSR is also not a fully constructed long trajectory: it is a lightweight **answer-invariant scaffold** whose purpose is to decouple planning structure from answer content. In other words, Yeo et al. asks whether hand-constructed full trajectories match emergent teacher traces, while we ask whether an answer-invariant scaffold reduces anchoring relative to unconstrained answer-conditioned reverse generation. These are different comparisons, so the conclusions are not contradictory. In fact, we view them as complementary: their result that fully hand-constructed long trajectories may be weaker than emergent teacher traces is compatible with our result that a lightweight structural scaffold reduces anchoring in the answer-visible reverse-CoT setting. We will add a discussion paragraph clarifying this distinction in the final version.

---

### Decision · Program_Chairs · 2026-04-30

**Decision:**

Accept (regular)

**Comment:**

The paper formalizes post-hoc rationalization in reverse Chain-of-Thought generation through a well-motivated three-level anchoring hierarchy and draws an insightful connection to Ironic Process Theory to explain why semantic suppression backfires. The proposed SSR method, which uses answer-invariant structural skeletons, consistently reduces anchoring across all levels, and SSR-D achieves strong downstream improvements with OOD generalization. Reviewers acknowledge the comprehensive experiments, the creative measurement framework, and the clear writing. Concerns about single-teacher-model bias, metric applicability to black-box models, and computational overhead were raised but adequately addressed during rebuttal, with cross-model validation and LLM-as-judge experiments strengthening the evidence.